# Forced Swimming-Induced Depressive-like Behavior and Anxiety Are Reduced by Chlorpheniramine via Suppression of Oxidative and Inflammatory Mediators and Activating the Nrf2-BDNF Signaling Pathway

Hasan S. Alamri [1], Rana Mufti [2], Deema Kamal Sabir [3], Abdulwahab A. Abuderman [4], Amal F. Dawood [5,*], Asmaa M. ShamsEldeen [6,7], Mohamed A. Haidara [6], Esma R. Isenovic [8] and Mahmoud H. El-Bidawy [4,6]

1 Department of Internal Medicine, College of Medicine, King Khalid University, P.O. Box 641, Abha 61421, Saudi Arabia; hsalamri@kku.edu.sa

2 Department of Clinical Sciences, College of Medicine, Princess Nourah bint Abdulrahman University, P.O. Box 84428, Riyadh 11671, Saudi Arabia; rsmufti@pnu.edu.sa

3 Department of Medical-Surgical Nursing, College of Nursing, Princess Nourah bint Abdulrahman University, P.O. Box 84428, Riyadh 11671, Saudi Arabia; dksaber@pnu.edu.sa

4 Department of Basic Medical Sciences, College of Medicine, Prince Sattam Bin Abdulaziz University, P.O. Box 11942, Al-Kharj 16278; Saudi Arabia; a.abuderman@psau.edu.sa (A.A.A.); m.elbidawy@psau.edu.sa (M.H.E.-B.)

5 Department of Basic Medical Sciences, College of Medicine, Princess Nourah bint Abdulrahman University, P.O. Box. 84428, Riyadh 11671, Saudi Arabia

6 Department of Physiology, Kasr Al-Aini Faculty of Medicine, Cairo University, Cairo 11566, Egypt; dr_asmaashams82@cu.edu.eg or asmaamshams.med@o6u.edu.eg (A.M.S.)

7 Department of Physiology, Faculty of Medicine, October 6 University, Cairo 11566, Egypt

8 Department of Radiobiology and Molecular Genetics, "VINČA" Institute of Nuclear Sciences-National Institute of the Republic of Serbia, University of Belgrade, 11000 Belgrade, Serbia; isenovic@yahoo.com

* Correspondence: afdawood@pnu.edu.sa

**Abstract:** The first-generation antihistamine chlorpheniramine (CPA) is believed to have both anxiolytic and antidepressant properties. The current study sought to assess the mechanisms behind the antidepressant and anxiolytic effects of CPA therapy concerning oxidative stress, inflammation, and nuclear factor p45 for erythroid 2-Brain-derived neurotrophic factor (Nrf2-BDNF) signaling pathway in forced swimming-induced depressive-like behavior and anxiety. Eighteen male Wistar rats (180–200 gm) rats were separated into three groups (n = 6): a stressed group (acute stress) that underwent the forced swimming test (FST) and a stressed group that received pretreatment with CPA (10 mg/kg body weight) for 3 weeks (CPA + acute stress). Animals were subsequently put through the following behavioral tests after undergoing a forced swim test (FST) for 5 min: an immobility test, open field test, and elevated plus maze test. Serum cortisol levels were measured when the rats were euthanized at the end of the experiments. Brain neurotransmitters (cortisol, serotonin, and noradrenaline), oxidative stress (SOD and MDA), inflammatory (IL-6 and IL-1) biomarkers, and the Nrf2-BDNF signaling pathway in the hippocampus and cerebral cortex tissues was determined. CPA prevented stress-induced increases in cortisol levels ($p < 0.0001$), decreased brain neurotransmitters, and increased oxidative stress and inflammation. CPA also upregulated the Nrf2-BDNF signaling pathway. Thus, CPA mitigates depressive-like behavior and anxiety by inhibiting oxidative stress and inflammation and upregulating the Nrf2-BDNF signaling pathway in the brain tissues.

**Keywords:** chlorpheniramine; forced swimming test; depressive-like behavior; anxiety; oxidative and inflammatory mediators; Nrf2-BDNF signaling pathway

## 1. Introduction

A variety of behavioral and cognitive changes, either active or passive, can be categorized as stress-coping mechanisms [1]. Following the animal coping mechanisms in

reaction to stress, researchers discovered a shift in behavior indicative of altered brain plasticity, neurotransmitter balance, cytokines, and altered blood cortisol levels [2,3].

Animal models of behavioral and/or emotional disorders become a very alluring subject for evaluating anxiety-like (open field test, novelty suppressed feeding, elevated plus maze, light/dark box, stress-induced hyperthermia) and depression-like behaviors [4].

The FST is one example of an animal behavior test that depends on conditioned place preference. Porsolt and his coworkers created FST to assess the therapeutic potential of antidepressant medications [5]. The FST can, however, be used to generate acute stress and investigate the neurobiology of stress management, which is pertinent to depressive-like behavior [6].

Serotonergic, dopaminergic, and noradrenergic systems have been demonstrated to play a role in the development of depressive-like behavior [7,8]. Drugs that increase serotonin, noradrenalin [9], and dopamine [10] have been found to lessen and restore function in depressed patients.

Cortisol is a marker that changes in both acute and chronic psychological stresses [11]. This is consistent with studies by scientists indicating that acute stress exposure raises serum ACTH and cortisol levels and affects immunological response [12].

Reactive oxygen species (ROS) generation is increased by oxidative stress, which also disrupts cellular structure and function [13]. The unregulated production of oxidants leads to oxidative stress, which affects cellular function and contributes to the growth of cancer, chronic disease, and toxicity [14]. According to reports, there is a connection between high ROS levels and several neurodegenerative disorders, including Alzheimer's and Parkinson's [15]. In particular, the adverse effects of oxidative offence can significantly impact the hippocampus [16]. Data indicated a connection between oxidative stress and mental disorders [17,18]. In certain investigations, the oxidative stress pathways were targeted as a potential therapeutic target [19].

According to data, excessive ROS generation leads to the synthesis of cytokines that promote inflammation [20]. Additionally, it was discovered that psychological stress altered immune function and raised serum levels of inflammatory cytokines [21] and the hippocampus, in particular, in the brain [22]. Depressive-like behavior and anxiety are strongly connected with elevated IL-6 levels [12,23].

It has been discovered that activation of the Nrf2 signaling pathway plays a significant role in suppressing oxidative damage [24]. Nrf2 controls the adaptive response to oxidants by acting as a xenobiotic-activated receptor [25]. Data showed that Nrf2 has anti-inflammatory properties. The NF-B pathway and the synthesis of pro-inflammatory cytokines are inhibited by Nrf2's prevention of inflammation [26]. According to Yao et al., Nrf2 activation boosts BDNF expression by lowering the expression of its transcriptional repressors, which results in effects similar to those of a fast-acting antidepressant [27].

Data from another study showed that continued antidepressant use has been shown to positively control the hippocampal expression of BDNF, which is adversely regulated by stresses [28]. Furthermore, depressive disorder in adults was associated with decreased BDNF [29].

The first-generation antihistaminic medication chlorpheniramine was examined in numerous trials for its anxiolytic-like effects [30]. It was discovered that CPA has an antidepressant effect in a mouse model of anxiety [31]. According to some studies, the administration of CPA boosts dopamine release in humans [32]. CPA controls emotional and behavioral processes in the rat brain by blocking serotonin uptake in the neuronal synapses [33].

We hypothesized that CPA functions as an antidepressant and an anxiolytic. However, not much research has been conducted on the mechanisms underlying its antidepressant and anxiolytic effects. This study's objectives were to assess the behavioral dysfunction brought on by forced swimming and to associate the behavioral consequences of CPA with oxidative stress, inflammation, and the Nrf2-BDNF signaling pathway.

## 2. Materials and Methods

### 2.1. Animals

In an animal facility, rats (male albino rats, 180–200 g) were housed in a clean animal room with a controlled temperature of 22 ± 2 °C and a relative humidity of 50–10%. They had unlimited access to food and water while being kept in cages with 12-h light/dark cycles.

### 2.2. Experimental Design

After a week of acclimation, 18 Wistar rats were evenly divided into three groups (n = 6 rats in each group). Three groups of animals were separated. Group 1 contained control normal rats receiving saline (i.p.) pretreatment for 2 weeks, Group 2 contained rats that underwent a forced swimming stress test after receiving saline (i.p.) pretreatment for 2 weeks, and Group 3 contained rats that underwent a forced swimming stress test after receiving CPA dissolved in saline (i.p.) at a dose of 10 mg/kg for 2 weeks after. All experimental protocols adhered to the guidelines of the Animal Welfare Committee of Universidad Complutense following European legislation (2010/63/EU). The current study was approved by the 6th October University Research Ethical Committee; 06U REC, (NO: PRE-Me-2103018).

All animals underwent behavioral tests after the experiment. Next, rats were given sodium phenobarbital anesthesia (40 mg/kg body weight), and blood was drawn via cardiac puncture into plain tubes for serum separation and cortisol measurement (schematic presentation showing all experimental steps conducted is shown in Figure 1).

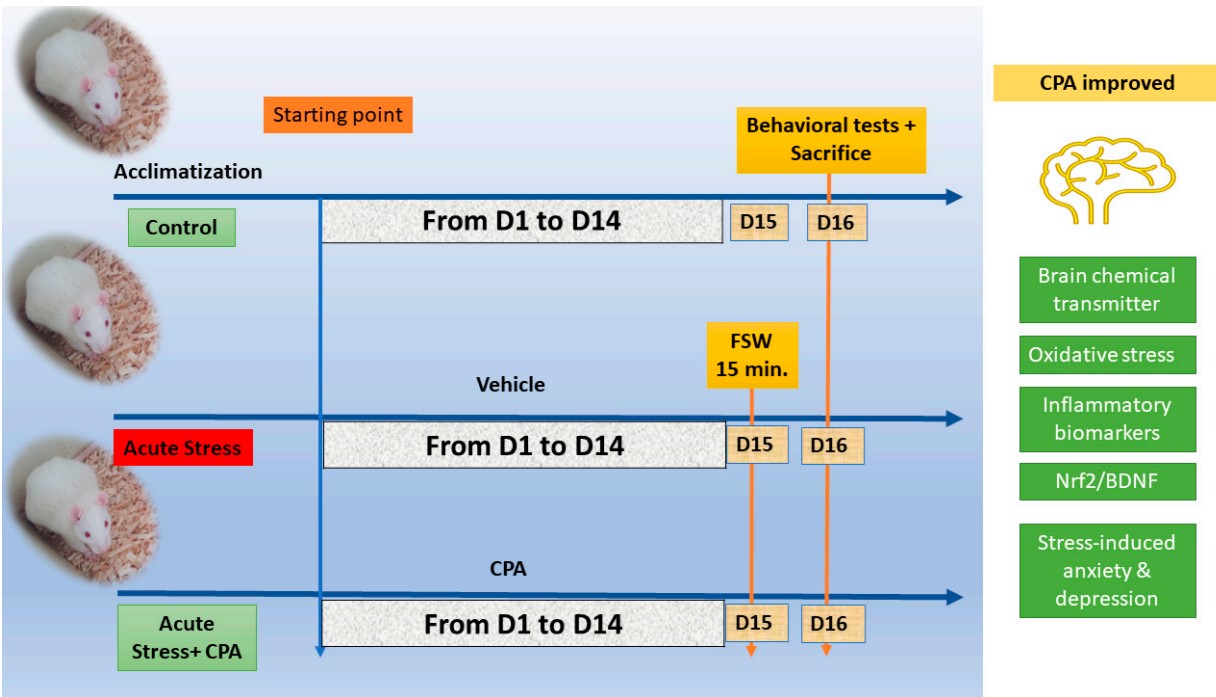

**Figure 1.** The proposed model for acute stress-induced depressive-like behavior and anxiety appears to be ameliorated by chlorpheniramine. CPA: chlorpheniramine; FSW: forced swimming; D1, D2, D15, and D16: day 1, day2, day 15, and day 16, respectively. Nrf2: Nuclear factor p45 for erythroid 2; Brain-derived neurotrophic factor: Brain-derived neurotrophic factor.

The animals were subsequently sacrificed via decapitation, and the brain tissues were harvested to assess the hippocampus MDA, SOD, IL-1β, and IL-6. Serotonin, dopamine, Nrf2, and BDNF levels were assessed in the cerebral cortex. At −80 degrees, brain tissue and serum were both preserved.

*2.3. Behavioral Tests*

The same researcher performed all behavioral tests between 8:00 a.m. and 1:00 p.m. Rats were given 30 min to adjust to the test environment before being tested. According to our previously published paper, all behavioral tests were conducted during a specific time frame, and each group was tested separately. Moreover, the order of the tested animal was chosen randomly within each group.

2.3.1. Forced Swim Test (FST) and Immobility Time

This test was designed to evaluate depressive-like, according to Porsolt et al. [5], but the depth of the water was deeper. Rats were placed in individual glass cylinders (50 cm tall × 20 cm in diameter and 30 cm deep) filled with water that was 23–25 °C to conduct swim sessions. The rats could not maintain themselves at the 30 cm water depth by touching the bottom with their feet, and only a few could do so with their tails. Two swimming sessions were held, the first lasting 15 min and the second lasting 5 min, separated by 24 h. When a rat stopped struggling and floated still in the water, making only the movements required to keep its head above water, it was deemed to be immobile. After each swim session, the rats were taken out of the cylinders, dried with paper towels, kept in heated enclosures for 15 min, and then put back into their original cages. Test sessions were videotaped and recorded for later scoring. This technique explains the forced swim test, which ultimately results in immobility time [34] and is thought to represent rodent depressive-like behavior [35].

2.3.2. Open Field Test

Rat anxiety-related behaviors were evaluated using OFT. The arena was divided into 25 squares, and the OFT was conducted using a square box (80 × 80 × 50 cm). The animals are brought to the testing room two hours before the examination on test day so they have time to get used to the setting. The center area was illuminated to 100 lx using an LED light source attached above the arena LED light source above the arena's center. Each rat underwent a 5-minute behavioral assessment while being watched by a camera that was positioned above the arena. In a 6-minute session, the length of time spent in the center, the number of line crossings made with all paws, the number of defecation boils, and grooming were all carefully counted. The equipment was cleaned with 10% ethanol between experiments to remove animal clues, and the light was kept at its lowest setting to prevent anxious behavior [36].

2.3.3. Elevated Plus Maze (EPM) Test

This experiment was carried out to assess the rats' anxiety-related behavior. Two open and two closed arms (50 × 10 × 40 cm) positioned 50 cm above the flat floor made up the construction of the maze. A camera positioned above the arena observed each rat's activity for five minutes after it was placed in the junction of the four arms of the EPM. The duration spent in the open arms, the number of open arms entries made, the frequency of head dipping, and the stretched-attend postures were all recorded [37].

*2.4. Biochemical Analysis*

2.4.1. Estimation of Serum Cortisol Levels

Using the cortisol (Cor) Rat ELISA Kit, Catalog #MBS453321, MyBioSource, Southern San Diego, CA, USA, serum cortisol levels were measured. All procedures were carried out following the manufacturer's instructions.

The cerebral cortex and hippocampus were separated and homogenized, and the supernatant was removed after the dissection of the brain tissues. Serotonin, dopamine, BDNF, and Nrf2 concentrations in the cerebral cortex supernatant were measured, while the hippocampal supernatant was used to estimate MDA, SOD, IL-1β, and IL-6.

### 2.4.2. Estimation of Serotonin and Noradrenaline Levels

Using an ELISA kit (Serotonin or Noradrenaline Research ELISA, catalogue numbers: BA E-5900 and BA E-5300, respectively, Labor Diagnostika Nord (LDN), Nordhorn, Germany), the total concentrations of serotonin and noradrenaline in cerebral cortex tissues were calculated. All procedures were carried out following the manufacturer's instructions.

### 2.4.3. Real-Time Polymerase Chain Reaction (RT-PCR) for Estimation of Gene Expression of BDNF and Nrf2

Brain tissue was processed using the miRNeasy Micro kit's standard methodology to extract total RNA (QIAGEN). Using a "high-capacity" cDNA reverse transcription kit (Applied Biosystems), 250 ng of total RNA was reverse-transcribed. All procedures were carried out following the manufacturer's recommendations. Using the SensiMix SYBR Hi-Rox Kit (Bioline; Meridian Life Science), 20 ng of the generated cDNA was amplified. The relative quantification of materials was performed using the [DELA][DELA]CT method. And, as internal controls, the outcomes were expressed in relation to the housekeeping gene beta-actin. The following primer sequences were utilized:

BDNF (sense); 5′-ACCATAAGGACGCGGACTTGT-3′; Nrf2 (sense); 5′-CCATGCCTTC TTCCACGAA-3′; beta-actin (sense); 5′-CCCATCTATGAGGGTTACGC-3′; and (antisense); 5′-TTTAATGTCACGCACGATTTC-3′ are examples of syllables.

### 2.4.4. Estimation of Biomarkers the Oxidative Stress Malondialdehyde (MDA) Levels and Superoxide Dismutase (SOD) and the Pro-Inflammatory Biomarkers IL-1 β and IL-6 in Brain Tissues

Brain specimens were homogenized in ice-cold saline and centrifuged for 15 min at $18,000 \times g$ (148C) to assess oxidative stress and inflammatory biomarkers. MDA was measured using the TBARS Assay Kit (Cayman Chemical Company, Ann Arbor, MI, USA; item number 10009055). The kit (Item No. 706002, Cayman Chemical Company, Ann Arbor, MI, USA) was used to quantify SOD. As per the manufacturer's instructions, IL-1 was measured using an ELISA kit (ClinMaxTM Human IL-1β ELISA Kit, Cat. No. CRB002-C01). Using an ELISA kit, the amount of IL-6 was measured (BIOTANG INC, Cat. No. RB1829, St, Lexington, MA, USA).

### *2.5. Statistical Analysis*

The data are expressed as mean ± standard deviation (S.D.). Data were processed and analyzed using GraphPad Prism (version 6). Data were double-checked for normality using the Shapiro–Wilk test and normality plots. The unpaired Student t-test was used to analyze differences between two groups in variable values that follow the normal distribution, while the Mann–Whitney test was used for non-normally distributed variables. One-way ANOVA followed by Tukey's post hoc test was used to analyze differences between three groups for normally distributed variables, while non-parametric Kruskal–Wallis was used for non-normally distributed variables. Pearson correlation statistical analysis was performed to analyze the correlation significance between two different parameters. Statistical significance was considered if $p \leq 0.05$ (Table 1).

**Table 1.** The ANOVA details (F statistic and degrees of freedom (df)).

|  |  | Sum of Squares | df | Mean Square | F | Overall *p* Value |
|---|---|---|---|---|---|---|
| MDA | Between Groups | 31,911.434 | 2 | 15,955.717 | 73.322 | <0.001 |
|  | Within Groups | 3264.182 | 15 | 217.612 |  |  |
|  | Total | 35,175.616 | 17 |  |  |  |
| SOD | Between Groups | 151.919 | 2 | 75.959 | 65.939 | <0.001 |
|  | Within Groups | 17.280 | 15 | 1.152 |  |  |
|  | Total | 169.198 | 17 |  |  |  |

**Table 1.** *Cont.*

|  |  | Sum of Squares | df | Mean Square | F | Overall *p* Value |
|---|---|---|---|---|---|---|
| IL-1 B | Between Groups | 44,693.818 | 2 | 22,346.909 | 247.961 | <0.001 |
|  | Within Groups | 1351.840 | 15 | 90.123 |  |  |
|  | Total | 46,045.658 | 17 |  |  |  |
| Noradrenaline | Between Groups | 17,965.671 | 2 | 8982.836 | 97.998 | <0.001 |
|  | Within Groups | 1374.947 | 15 | 91.663 |  |  |
|  | Total | 19,340.618 | 17 |  |  |  |
| Cortisol | Between Groups | 199.564 | 2 | 99.782 | 24.530 | <0.001 |
|  | Within Groups | 61.016 | 15 | 4.068 |  |  |
|  | Total | 260.580 | 17 |  |  |  |
| BDNF | Between Groups | 123,369.854 | 2 | 61,684.927 | 111.713 | <0.001 |
|  | Within Groups | 8282.623 | 15 | 552.175 |  |  |
|  | Total | 131,652.478 | 17 |  |  |  |
| No of line crossings (open field test) | Between Groups | 3164.778 | 2 | 1582.389 | 52.109 | <0.001 |
|  | Within Groups | 455.500 | 15 | 30.367 |  |  |
|  | Total | 3620.278 | 17 |  |  |  |
| Time spent in open arms (S) (elevated plus maze test) | Between Groups | 2362.333 | 2 | 1181.167 | 155.190 | <0.001 |
|  | Within Groups | 114.167 | 15 | 7.611 |  |  |
|  | Total | 2476.500 | 17 |  |  |  |
| Head dipping (elevated plus maze test) | Between Groups | 296.333 | 2 | 148.167 | 43.016 | <0.001 |
|  | Within Groups | 51.667 | 15 | 3.444 |  |  |
|  | Total | 348.000 | 17 |  |  |  |
| Stretching (elevated plus maze test) | Between Groups | 50.333 | 2 | 25.167 | 31.027 | <0.001 |
|  | Within Groups | 12.167 | 15 | 0.811 |  |  |
|  | Total | 62.500 | 17 |  |  |  |

## 3. Results

### 3.1. CPA Protects against Stress-Induced Depressive-like Behavior

The FST is the most popular behavioral test for assessing drugs to operate as an antidepressant [38]. We assessed the effect of PCA therapy on the immobility latency of rats subjected to FST based on its excellent predictive validity. When compared to the AS group, CPA significantly decreased immobility time and, consequently, depressive-like behavior ($p \leq 0.0001$) (Figure 2A). We also checked the serum cortisol levels, which are known to rise in depressive states (Figure 2B). Significantly raised serum cortisol levels in the As group compared to the control ($p \leq 0.0001$) support the idea that acute stress is linked to higher cortisol levels. Compared to acute stress, the administration of CPA significantly lowers serum cortisol levels, bringing them back to normal (insignificant). Furthermore, we demonstrated that AS significantly reduced serotonin ($p \leq 0.0001$) (Figure 2C) and noradrenaline ($p \leq 0.0001$) (Figure 2D) compared to the control, indicating that exposure to acute stress and depressive-like behavior is associated with modulation of brain neurotransmitters. Administration of CPA significantly raises neurotransmitter levels (not significant) compared to acute stress and returns them to control levels.

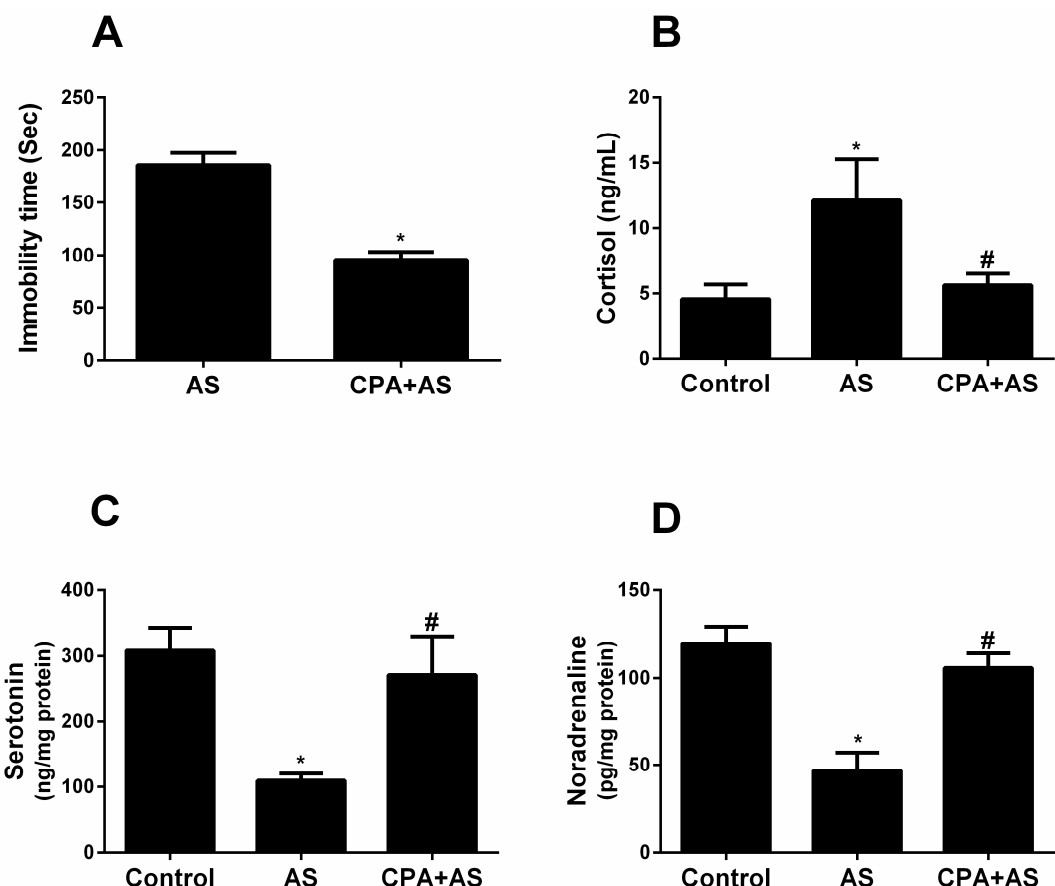

**Figure 2.** Chlorpheniramine (CPA) decreases immobility time as detected by a forced swimming test that assesses depressive-like behavior $p \leq 0.0001$ (**A**). CPA increased cortisol levels (ng/mL) in serum ($p \leq 0.0001$) and (**B**) serotonin (ng/mg protein), ($p \leq 0.0001$) (**C**), noradrenaline (pg/mg protein) ($p \leq 0.0001$) (**D**) in cerebral cortex tissues of acutely stressed (AS) group at the end of the experiment. The findings are the mean $\pm$ SD, n = 6. * $p$ when compared to the controls; # $p$ when CPA + AS is compared to the AS group.

### 3.2. CPA Protects against Stress-Induced Anxiety Using an Open Field Test

The following variables were measured during the open field Test (Figure 3A–D):

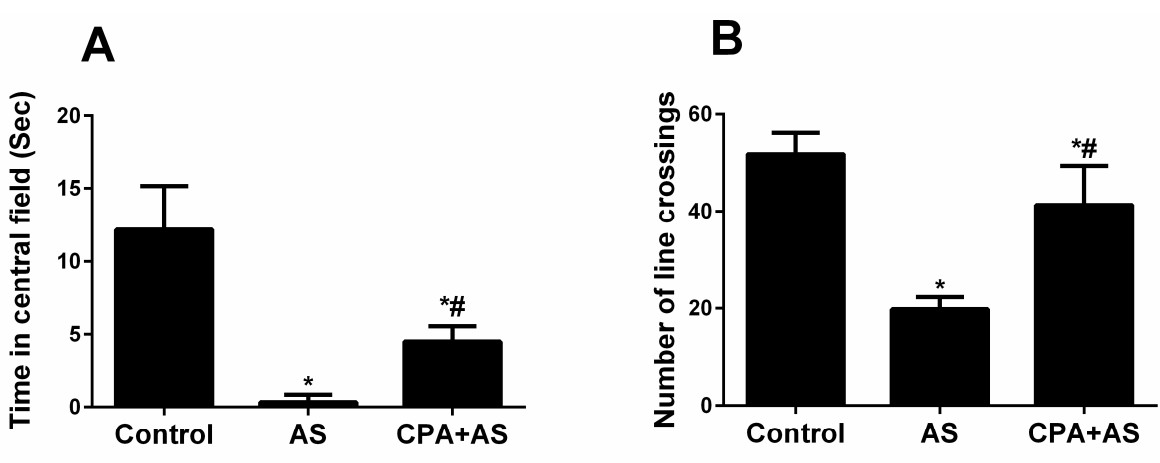

**Figure 3.** *Cont.*

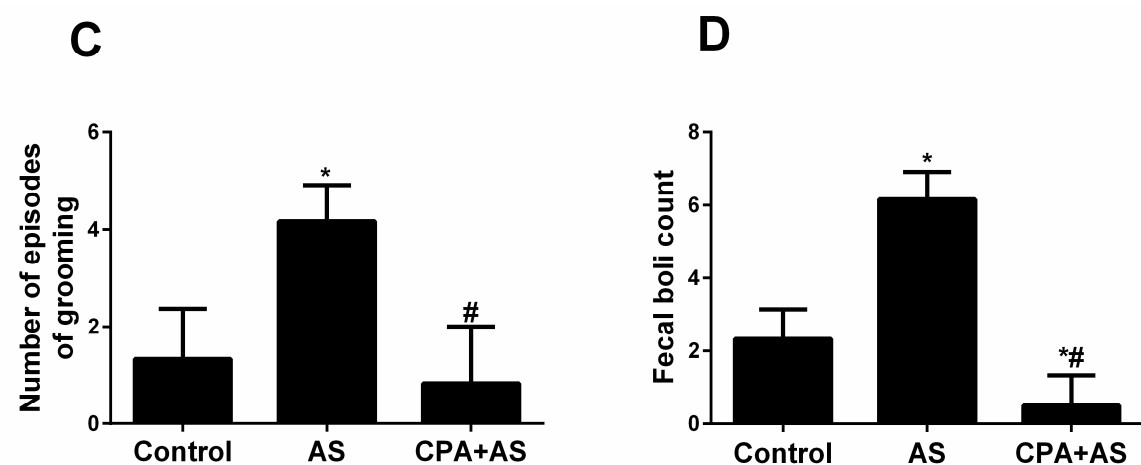

**Figure 3.** Data obtained from open field test. Chlorpheniramine (CPA) improved anxiety-related behavior detected by increased time spent in the central field ($p \leq 0.0001$) (**A**), increased locomotor activity confirmed by the number of crossing lines ($p \leq 0.0001$) (**B**), decreased number of grooming episodes ($p \leq 0.0005$) (**C**) and decreased count of the fecal boil ($p \leq 0.0001$) (**D**) in acute stressed (AS) group at the end of the experiment. The findings are the mean ± SD, n = 6. * $p < 0.05$ when compared to the controls, # $p < 0.05$ when CPA + AS is compared to the AS group.

### 3.2.1. Time Spent in the Field's Centre

Figure 3A revealed that all groups spent much less time in the field's center than the control group. Data revealed that although the CPA + AS group's decline was statistically significant compared to the AS group's decline ($p \leq 0.0001$), it did not eventually reach the control level.

### 3.2.2. Number of Line Crossings

Figure 3B demonstrated a significant decrease in the number of crossings in the AS group compared to the control, which was significantly enhanced by the administration of CPA ($p \leq 0.0001$), although it did not recover to control levels.

### 3.2.3. Number of Grooming Episodes

According to Figure 3C, there is a substantial increase in grooming in the AS group ($p \leq 0.0005$). This rise was reduced when CPA was administered but did not return to control levels.

### 3.2.4. Fecal Boli Count

Figure 3D demonstrated a significant rise in fecal boli count in the AS group ($p \leq 0.0001$), which was subsequently decreased by the administration of CPA but did not return to the control level.

The findings above demonstrated that CPA offers a defense against anxiety brought on by acute stress.

### 3.3. CPA Protects against Stress-Induced Anxiety Using an Elevated Plus Maize Test (EPMT)

During the behavioral studies, anxiety was also evaluated using an elevated plus maze test (Figure 4A–D), following the guidelines outlined in the preceding subsection. The following variables in the EPMT were measured:

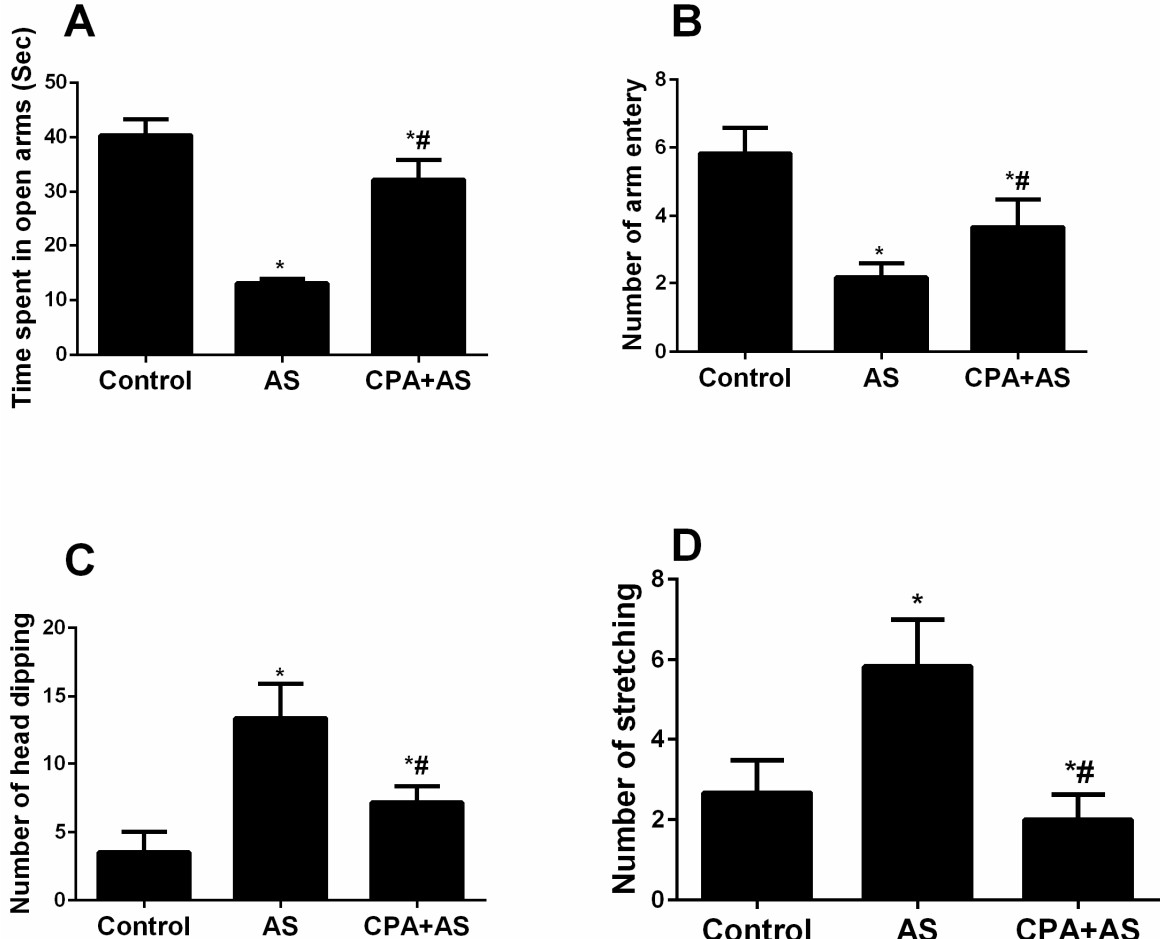

**Figure 4.** Data obtained from elevated plus maze test. Chlorpheniramine (CPA) improved anxiety-related behavior detected by increasing time spent in the open arm ($p \leq 0.0001$) (**A**), number of arm entries ($p \leq 0.0001$) (**B**), number of head dipping ($p \leq 0.0001$) (**C**) and number of stretching ($p \leq 0.0001$) (**D**) in acute stressed (AS) group at the end of the experiment. The findings are the mean $\pm$ SD, n = 6. * $p < 0.05$ when compared to the controls; # $p < 0.05$ when CPA + AS is compared to the AS group.

### 3.3.1. Time Spent in the Open Arms

Figure 4A demonstrated that, compared to the control group, all groups' time spent in the open arms was significantly lower. Data revealed that although the CPA + AS group's increase compared to the AS group was significant ($p \leq 0.0001$), it did not return to the control level.

### 3.3.2. Number of Arm Entries

According to Figure 4B, all groups' numbers of arm entries were significantly lower than those in the control group. Although the CPA + AS group showed a significant increase compared to the AS group ($p \leq 0.0001$), it is still significant compared to the control group.

### 3.3.3. Number of Head Dipping

The frequency of head dipping was considerably higher in all groups as compared to the control group, as seen in Figure 4C. Although the difference between the CPA + AS and AS groups was significantly reduced ($p \leq 0.0001$), it is still significant compared to the control group.

### 3.3.4. Stretched-Attend Postures

Stretched-attend postures were substantially more frequent in all groups compared to the control group, as shown in Figure 4D. Although the difference between the CPA + AS and AS groups was significantly reduced ($p \leq 0.0001$), it is still significant compared to the control group.

The EPMT results mentioned above indicate that CPA guards against anxiety brought on by acute stress.

### 3.4. CPA Attenuated Stress-Induced Oxidative Stress and Inflammatory Biomarkers in Hippocampal Tissues

Our findings demonstrated that AS increased oxidative stress through a significant rise in MDA ($p \leq 0.0001$) (Figure 5A) and a decrease in SOD ($p \leq 0.0001$) (Figure 5B) in hippocampal tissues when compared to the control. Compared to acute stress, the administration of CPA significantly reduces MDA ($p \leq 0.0001$) and raises SOD levels, but neither effect returns the levels to those of the control.

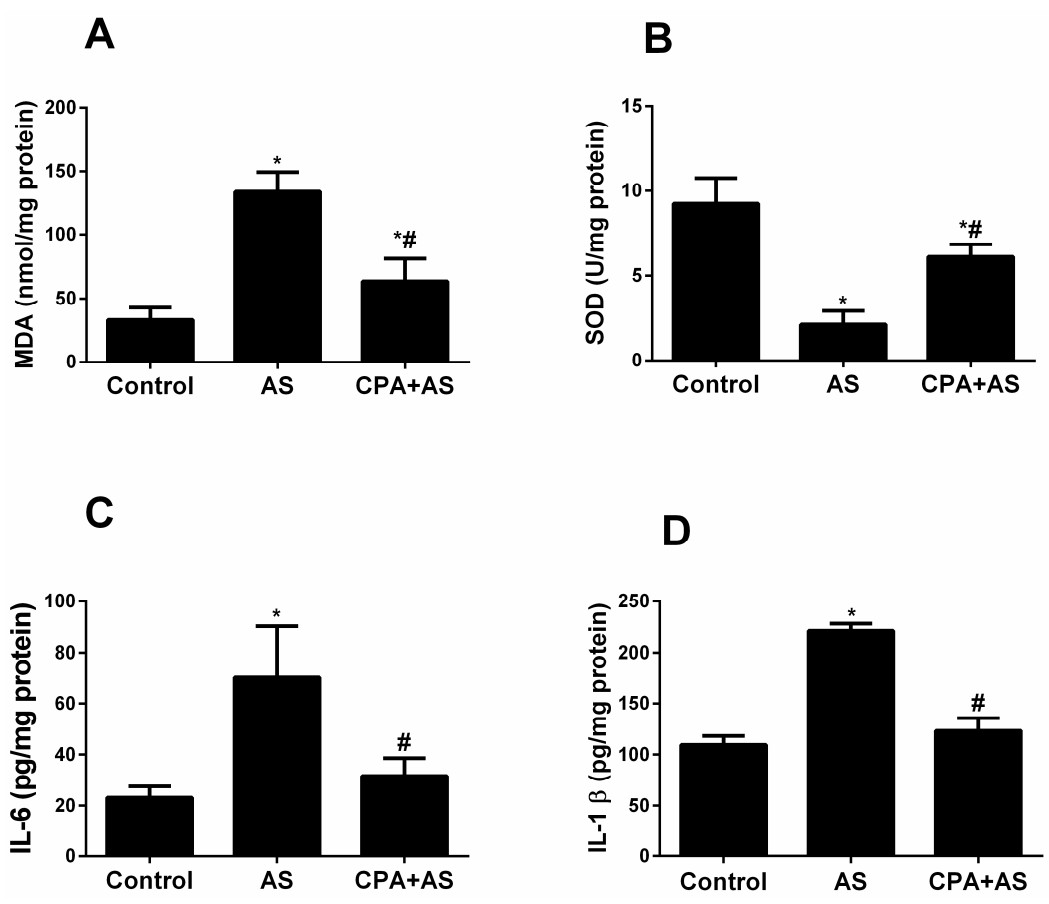

**Figure 5.** Biochemical parameters in the hippocampal tissues (oxidative stress and pro-inflammatory biomarkers). CPA decreases both oxidative stress (decreased MDA ($p \leq 0.0001$) (**A**), and increased SOD ($p \leq 0.0001$) (**B**)) and pro-inflammatory (decreased both IL-6 ($p \leq 0.0001$) (**C**), and IL-1β ($p \leq 0.0001$) (**D**)) biomarkers of the acute stressed (AS) group at the end of the experiment. The findings are the mean ± SD, n = 6. * $p < 0.05$ when compared to the controls; # $p < 0.05$ when CPA + AS is compared to the AS group.

Additionally, we demonstrated that AS elevated inflammatory biomarkers by demonstrating a significant increase in IL-6 ($p \leq 0.0001$) (Figure 5C) and IL-1β ($p \leq 0.0001$) (Figure 5D) levels compared to control. When CPA is administered, IL-6 ($p \leq 0.0002$) and IL-1B ($p \leq 0.0001$) levels are significantly lower than during acute stress and return to control values.

### 3.5. CPA Attenuated Stress-Induced Decrease in Nrf2 and BDNF in Cerebral Cortex Tissues

Our findings show that AS significantly lowers Nrf2 ($p \leq 0.0001$) and BDNF ($p$ 0.0001) levels in cerebral cortical tissues compared to the control. When compared to acute stress, the administration of CPA significantly increases Nrf2 levels ($p \leq 0.0001$), though it is still significant compared to the control ($p = 0.0303$). Compared to acute stress, CPA likewise dramatically raises BDNF ($p \leq 0.0001$), but it lowers BDNF back to control levels (Figure 6A,B).

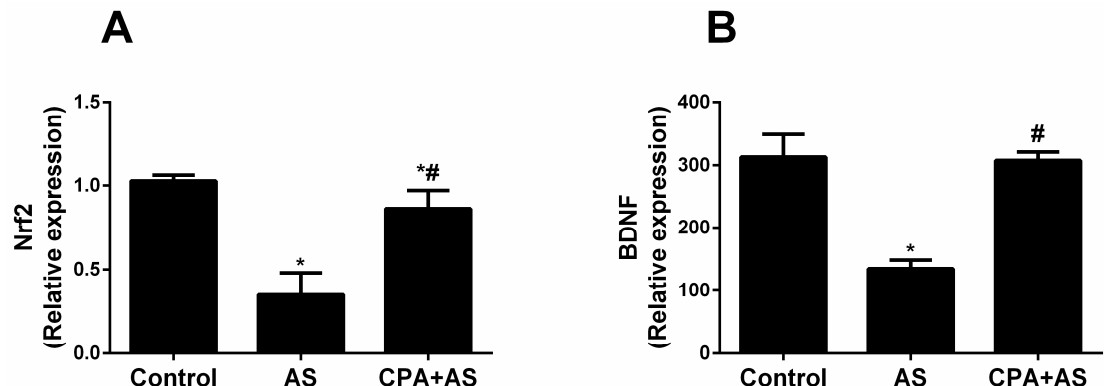

**Figure 6.** Biochemical parameters in the cerebral cortical tissues (relative expression levels of Nrf2 (**A**), and BDNF (**B**). CPA upregulated the Nrf2/BDNF signaling pathway of the acute stressed (AS) group at the end of the experiment. The findings are the mean $\pm$ SD, n = 6. * $p \leq 0.0001$ when compared to the controls; # $p \leq 0.0001$ when CPA + AS is compared to the AS group.

### 3.6. Correlation between Anxiety-Related Behavior and Changes in Brain Transmitters, Inflammatory Biomarkers, and Nrf2-BDNF Signaling Pathway

We determined the correlation between the number of line crossings as being representative of anxiety-related behavior and changes in brain transmitters, inflammation, and the Nrf2-BDNF signaling pathway. This links anxiety disorders with the biomarkers of brain injury, and it further supports the pleiotropic effects of CPA. The number of line crossings displayed a significant ($p < 0.0001$) negative correlation with IL-1 β (r = −0.895) (Figure 7A) and a positive correlation with serotonin (r = 0.842) (Figure 7B), Nrf2 (r = 0.891) (Figure 7C), and BDNF (r = 0.884) (Figure 7D).

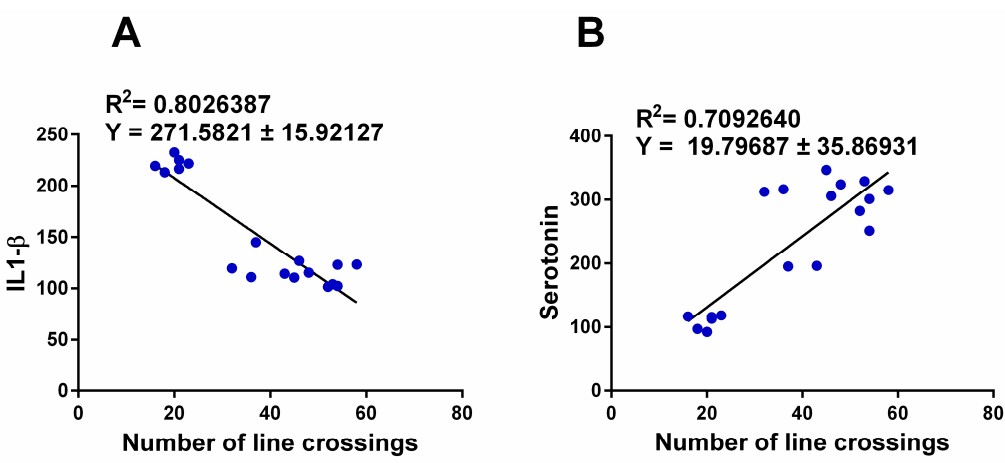

**Figure 7.** *Cont.*

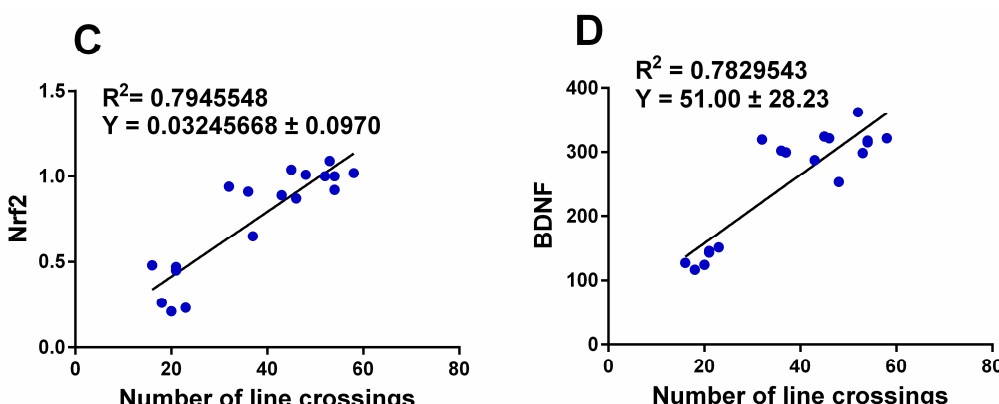

**Figure 7.** Correlation between the number of line crossings as being representative of anxiety-related behavior and IL-1 β (**A**), brain neurotransmitters serotonin (**B**), and Nrf2 (**C**)-BDNF (**D**) signaling pathway. All rats (n = 6 per group) are shown.

## 4. Discussion

This article investigated the induction of depressive-like behavior and anxiety with and without the combination of the antihistaminic drug chlorpheniramine in a rat model of the disease. We modelled this disease to test the hypothesis that CPA can ameliorate depressive-like behavior and anxiety induced by acute stress, associated with the augmentation of brain chemical transmitters, enhancement of the novel Nrf2-BDNF signaling pathway, and inhibition of biomarkers of oxidative stress and inflammation (Figure 1). Here, we report that the induction of depressive-like behavior and anxiety via the FST caused a significant decrease in brain chemical transmitters in the cerebral cortex (serotonin and noradrenaline). This was associated with the inhibition of hippocampal tissue levels of the antioxidant SOD and augmentation of the oxidative stress biomarker (MDA), inflammation biomarkers (IL-1β and IL-6), downregulation of the Nrf2-BDNF signaling pathway, and dysregulation of behavioral tests, which appeared to be protected by CPA (Figures 2–6). In addition, using the data obtained from the three animal groups, a significant correlation was observed between the number of line crossings as being representative of anxiety-related behavior inflammatory biomarkers (IL-6, IL-1 β) and the Nrf-2-BDNF signaling pathway (Figure 7), which further confirms that CPA is a beneficial pleiotropic medicine to treat acute stress-induced anxiety-related behavior. Therefore, our data support our working hypothesis mentioned above.

We measured immobility time in the Porsolt FWT to gauge behavioral despair associated with depressive-like behavior [39]. Our results indicated depressive-like behavior in our animal model as it is associated with increased mobility time (Figure 2A). However, Cryan et al. recommended that the modified rat FST become accepted as an improved method for characterizing the effects of antidepressant drugs and also for studying the neural substrates underlying their behavioral effects [40].

Additionally, our research demonstrates that our stressed model had elevated serum cortisol levels, which are known to rise during depressive episodes [41]. According to our behavioral tests, the administration of CPA reduced depressive-like behavior by preventing the stress-induced rise in blood cortisol (Figure 2B). Given that SSRIs and CPA share structural similarities, serotonergic activity may be one explanation for the antidepressant effects of CPA. (29). Another theory is that CPA acts as an antidepressant by increasing neurotransmitters other than serotonin, like norepinephrine (Figure 2B), as demonstrated by our findings. This aligns with a former study [31] that showed the antidepressant effects of CPA in rodents when given before the FST. We demonstrated that the FST causes anxiety using the elevated plus maze test (Figure 4A–D) and the open field test (Figure 3A–D). Acute stress reduces the time spent in the central field and the number of crossing lines in the open field test, but it increases grooming and the number of boli. Animals that were

evaluated in high plus maize displayed less time in open arms and arm entries as well as more head dipping and stretching. Our findings are consistent with the information gathered by Amin et al. [42], who documented animal anxiety following FST. Following three weeks of administration to rats, the current study found that CPA had anxiolytic properties. At various doses, Serafim et al. showed that CPA has anxiogenic effects in mice [43]. Since acetylcholine is well known to be associated with anxiety-like behaviors in rodents, the authors hypothesized that CPA functions as an H1 receptor antagonist that inhibits acetylcholine release from the ventral striatum [44]. On the other hand, other research revealed that CPA had anxiolytic effects in animals [45]. It has been proposed that the regulation of the serotonergic system may be responsible for the anxiolytic effects of CPA [46] ROS levels rise in several organs, including the hippocampus, in response to acute stress [47]. Our findings demonstrated a correlation between elevated oxidative stress and anxiety caused by FST and demonstrated via behavioral tests (Figure 5A,B). Stressed rats were given a three-week pretreatment with CPA (10 mg/kg body weight), which showed the antioxidant benefits of CPA by lowering MDA and raising SOD levels in hippocampus tissues. This is consistent with earlier research that indicated CPA reduced oxidative stress [48].

We demonstrated that SOD activity was reduced after acute stress induction (Figure 5A,B), whereas CPA dramatically increased SOD activity compared to the stressed group. The role of oxidative stress in mood disorders has recently drawn more attention. For instance, after acute stress exposure, SOD mRNA levels increased [49]. In contrast, a different study showed that mild chronic stressful events in mice reduced SOD activity in the cortex and hippocampus [50]. This may indicate a potential role for the antioxidant enzyme in acute stress. Data showed that stress is linked to higher MDA, which is consistent with our findings [51].

According to previous studies, ROS overproduction resulted in cytokine production that promotes inflammation [52]. Increased hippocampal tissues and blood pro-inflammatory cytokines have been linked to stress [53].

Our findings demonstrated that acute stress was linked to elevated inflammatory biomarkers IL-6 and IL1B (Figure 5C,D. This is consistent with findings that indicated IL-6 and IL1B levels were elevated in depressive-like behavior and anxiety [54]. IL-1β levels in the dorsal hippocampus after the stress-enhanced fear model were similarly shown to be higher in the data [55]. Our findings demonstrated that administering CPA lowers pro-inflammatory cytokine levels in the hippocampus tissues, which is related to reduced behavioral dysfunction, as demonstrated by our behavioral tests.

In conclusion, to the best of our knowledge, this is the first study to link the anxiolytic and antidepressant properties of CPA to the control of oxidative stress and inflammation in conjunction with the activation of the Nrf2-BDNF signaling pathway in a rat model of acute stress. This early finding offers original hypotheses for the next research on depressive-like behavior and anxiety. Therefore, more research is needed to clarify the potential application in clinical practice.

## 5. Study Limitation

Despite our groundbreaking results, this study has some drawbacks. The FST is not an animal model of depression. It is a model that can detect molecules that may possess antidepressant activity. In other words, it is a pharmacological predictive model with significant translational shortcomings. Only a clinical study can decide if a molecule is an antidepressant or not. In other words, the FST only indicates if a molecule has an antidepressant effect. When we extend our research, we will induce stress and consider FST as a test, not an inducer of depression.

It is strongly suggested that more studies using a dose–response curve be conducted. Future research may be successful in identifying more pathways that control inflammatory biomarkers in various parts of the brain. Also, measurements of Nrf2 and BDNF utilizing Western blot to measure the actual protein level would be considered when we extend our

research. Last but not least, the ability of CPA to guard against stress-related depressive-like behavior and anxiety was the main focus of this investigation. It could be more instructive to look at this impact on brain function and all other measured markers over a more extended period. Additionally, only a potential protective effect of CPA against stress-related brain damage was demonstrated in this study. The ability of CPA to prevent stress-related harm to other organs, such as the heart, liver, lungs, and kidneys, was not examined in this study but will be in subsequent ones.

Endothelial dysfunction (ED) has been connected with various clinical disorders, including depression and cardiovascular risk [56–58]. Furthermore, bipolar depression and increased vascular endothelial growth factor (VEGF) levels have frequently been linked [59]. The profile of circulating endothelium damage markers identified in the forced swimming-induced behavioral impairment and the relationship between the behavioral effects of CPA and nitric oxide (NO) and VEGF levels should be investigated in further studies.

**Author Contributions:** Conceptualization: M.A.H. and H.S.A.; Methodology: A.M.S., R.M. and D.K.S.; Investigation: A.F.D., M.H.E.-B., A.M.S. and R.M.; Formal Analysis: M.A.H., A.A.A., A.F.D., M.H.E.-B. and H.S.A. Data Curation: A.A.A., E.R.I., A.M.S., R.M., D.K.S. and M.H.E.-B.; Funding Acquisition: A.F.D.; Supervision: M.A.H. and H.S.A.; Writing—Original Draft Preparation: M.A.H., H.S.A. and A.F.D.; Writing—Review and Editing: M.A.H. and E.R.I. All authors have read and agreed to the published version of the manuscript.

**Funding:** This work was funded by Princess Norah bint Abdulrahman University Researchers Supporting Project number (PNURSP2023R110), Princess Nourah bint Abdulrahman University, P.O. Box 84428, Riyadh 11671, Saudi Arabia.

**Institutional Review Board Statement:** The study was conducted following the Declaration of Helsinki and approved by the Research Ethics Committee of 6th October University; 06U REC, (NO: PRE-Me-2103018).

**Informed Consent Statement:** Not applicable.

**Data Availability Statement:** The data supporting this study's findings are available on request from the corresponding author.

**Acknowledgments:** We thank Yasmeen Haidara, Psychologist, American University Cairo (AUC), Egypt, for editing and proofreading the manuscript.

**Conflicts of Interest:** The authors declare no conflict of interest.

**Sample Availability:** Samples of the compounds are available on request from the corresponding author.

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
