# Peer review of "Forced Swimming-Induced Depressive-like Behavior and Anxiety Are Reduced by Chlorpheniramine via Suppression of Oxidative and Inflammatory Mediators and Activating the Nrf2-BDNF Signaling Pathway"

_cimb, doi:10.3390/cimb45080407_

Round 1
Reviewer 1 Report
The paper is interesting and well written. The authors analyzed how chlorpheniramine impacts on oxidative stress and inflmmation. The endpoints of the study are well defined and described. The introduction and methods are adequante and coerent with the objectives. Results and discussion are well described. I suggest to briefly discuss the role of VEGF in endothelial dysfunction and if chlorpherinamine may have a positive impact on VEGF as TNF alpha inhibitors (see and add as references papers by Murdaca et al concerning endothelial dysfunction and VEGF, and concerning the efficacy of TNF alpha inhibitors on VEGF)
Minor engligh editing
Author Response
Manuscript ID: cimb-2478951.
Submission Title: Forced swimming-induced- depression and anxiety are reduced by chlorpheniramine by suppressing oxidative and inflammatory mediators and activating the Nrf2-BDNF signaling pathway.
Revised Title: Forced swimming induced depressive-like behavior and anxiety are reduced by chlorpheniramine via suppression of oxidative and inflammatory mediators and activating the Nrf2-BDNF signaling pathway
Authors: Hasan S. Alamri, Rana Mufti, Deema Kamal Sabir, Abdulwahab A. Abuderman, Amal F. Dawood, Asmaa M. ShamsEldeen, Mohamed A. Haidara, Esma R. Isenovic and Mahmoud H El-Bidawy
We express our sincere gratitude for the work and time the Editor and Reviewers committed to helping us improve our manuscript. Following the reviewers' recommendation, we have corrected all pointed issues. We revised the manuscript's parts according to the reviewers' suggestions and provided the requested additional information. The changes in the manuscript text are highlighted in yellow. We believe that we have addressed all of the Reviewers' concerns, and we hope that the revised version of the manuscript is suitable for publication in the Current Issues in Molecular Biology.
RESPONSES TO THE EDITORS' COMMENTS
COMMENT. Please modify references in the text where reference numbers should be in square brackets and placed before punctuations.
RESPONSE: We appreciate the editor's input and apologize for deviating from the Journal's approach. In response to this comment, reference numbers are enclosed in square brackets and placed before punctuation in the updated version of the MS.
RESPONSES TO THE REVIEWERS' COMMENTS
Reviewer #1
COMMENT 1: The paper is interesting and well written. The authors analyzed how chlorpheniramine impacts on oxidative stress and inflammation. The endpoints of the study are well defined and described. The introduction and methods are adequate and coherent with the objectives. Results and discussion are well described..
RESPONSE: We thank the Reviewer for his encouraging comments.
COMMENT 2: I suggest to briefly discuss the role of VEGF in endothelial dysfunction and if chlorpherinamine may have a positive impact on VEGF as TNF alpha inhibitor (see and add as references papers by Murdaca et al concerning endothelial dysfunction and VEGF, and concerning the efficacy of TNF alpha inhibitors on VEGF).
RESPONSE: We appreciate the Reviewer's valuable comments and suggested references. In response to this comment, we have included a more comprehensive discussion of VEGF's role in endothelial dysfunction. The added text now reads as follows:
"Endothelial dysfunction is characterized by reducing the bioavailability of vasodilators, particularly nitric oxide (NO). In addition, endothelial dysfunction comprises a specific state of endothelial activation characterized by a proinflammatory milieu. One of the potential mechanisms explaining the high risk of cardiovascular diseases in people with depression may be related to endothelial dysfunction (Murdaca et al., 2013).
Vascular endothelial growth factor (VEGF) primarily exerts its effect by producing vasodilating mediators. VEGF signaling increases NO production. However, in the neurotrophic model of depression, stress is d with low VEGF levels; the latter may cause atrophy of limbic structures resulting in depressive symptoms (Castillo et al., 2020).
The constitutional expression of COX-2 in the brain, as well as the growing recognition of the role of inflammation in psychiatric disease, support the growth potential of CBX in psychiatric diseases, as found in other studies such as MDD (Na et al., 2016) and schizophrenia (Zheng et al., 2017; Müller et al., 2010). According to another study (Kuwano et al., 2004), inflammatory cytokines such as IL-1 and TNF frequently increase COX-2 mRNA and protein expression in many human cell types. COX-2 selective drugs effectively decreased angiogenesis induced by IL-1, but not a VEGF receptor tyrosine kinase inhibitor.COX-2 selective drugs somewhat inhibited VEGF-induced angiogenesis (Kuwano et al., 2004)."
References:
Giuseppe Murdaca, Francesca Spanò, Paola Cagnati & Francesco Puppo (2013) Free radicals and endothelial dysfunction: Potential positive effects of TNF-α inhibitors, Redox Report, 18:3, 95-99, DOI: 10.1179/1351000213Y.0000000046
Castillo MFR, Cohen A, Edberg D, Hoppensteadt D, Fareed J, Martin B, Halaris A. Vascular endothelial growth factor in bipolar depression: A potential biomarker for diagnosis and treatment outcome prediction. Psychiatry Res. 2020 Feb;284:112781. doi: 10.1016/j.psychres.2020.112781. Epub 2020 January 11. PMID: 31986357.
Na, K., Lee, K.J., Lee, J.S., Cho, Y.S., Jung, H., 2016. Corrigendum to "Efficacy of adjunctive celecoxib treatment for patients with major depressive disorder: A meta-analysis" [Prog. Neuro-Psychopharmacol. Biol. Psychiatry 48 (January 3 2014) 79–85]. Progr. Neuropsychopharmacol. Biol. Psychiatry 66, 136. https://doi.org/10. 1016/j.pnpbp.2013.09.006.
Müller, N., Krause, D., Dehning, S., Musil, R., Schennach-Wolff, R., Obermeier, M., Möller, H.J., Klauss, V., Schwarz, M.J., Riedel, M, 2010. Celecoxib treatment in an early stage of schizophrenia: results of a randomized, double-blind, placebo-controlled trial of celecoxib augmentation of amisulpride treatment. Schizophr. Res. 121, 118–124. https://doi.org/10.1016/j.schres.2010.04.01
Zheng, W., Cai, D., Yang, X., Ungvari, G., Ng, C., Muller, N., Ning, Y., Xiang, Y., 2017. Adjunctive celecoxib for schizophrenia: a meta-analysis of randomized, double-blind, placebo-controlled trials. J. Psychiatr. Res. 92, 139–146. https://doi.org/10.1016/j. jpsychires.2017.04.004
Kuwano, T., Nakao, S., Yamamoto, H., Tsuneyoshi, M., Yamamoto, T., Kuwano, M., Ono, M., 2004. Cyclooxygenase 2 is a key enzyme for inflammatory cytokine-induced angiogenesis. FASEB J. 18, 300–310. https://doi.org/10.1096/fj.03-0473com

Reviewer 2 Report
The manuscript by Alamri et al., provides an important information on chlorpheniramine (CPA) that as it reported has antidepressant-like and anxiolytic-like activity in acutely stressed rats. The results are sounds, but the manuscript written sloppily, and I have some comments that need to be addressed, as follows:
Majors
1) Since authors indicate the sample size as 6 animals in each group I have a serious concern regarding the statistical power that complicates the ability to draw appropriate interpretations from the results. Did the power analysis was performed? Degrees of freedom for ANOVAs are not indicated anywhere in the results section. So, the next question is: whether outliers have been excluded? For comparison of two groups for immobility time the t-test is more suitable.
2) In the “materials and methods” section, the scheme of the experiment is required. So, the figure 8 is more appropriate in the paragraph 2.2. From the experimental design description it is not clear when exactly and in what sequence the OF and EPM tests were carried out. Were they carried out on days 16-20? From the description, I got the impression that both tests were performed on the same day. If this is true, then this is a significant drawback.
3) Throughout the text, the authors use the term "depression". This is incorrect if we are talking about rodents. We can only use terms “depressive-like behavior” or “behavioral despair”.
4) The authors assessed only BDNF mRNA level, but in fact this may not reflect the changes in BDNF protein level. For example, proBDNF level may be increased, while the level of mature BDNF may not. So, this needs to be discussed and mentioned among the study limitations.
5) The statement “downregulation of the Nrf2-BDNF signaling pathway” in the discussion is too sounds. Actually only two components of this signaling pathway have been analyzed and each of them could changed independently. Considering a plethora of factors affecting BDNF expression, such a scenario is quite possible.
Minors
1) I assume that the distribution was normal, which led to the choice of parametric ANOVA test, however, in the "statistical analysis" section, it is necessary to indicate whether a test for normality was carried out.
2) In the figure legends as well as in the abstract the sample number indicated as n = 8. Obviously, this is mistake.
3) P values and indication on the graphs are inconsistent. For example, for immobility time indicated p ≤ 0.0001 while on the figure 1A placed (*) instead (***) as widely accepted for p ≤ 0.0001. Such discrepancies are throughout the text. Also I recommend to denote control vs. CPA+AS and AS vs. CPA+AS comparisons with different symbols (i.e. * and #). This should facilitate the visual perception of the material.
4) Subheading in the paragraph 3.6 obviously from any other paper.
5) Figure 2C: correctly as “number of episodes of grooming”
6) “The same researcher performed all behavioral tests…” in the paragraph 2.3: what does it mean? This sentence needs to be made clear.
7) The sentence “mental illnesses, including Alzheimer’s and Parkinson’s” is incorrect. Mentioned disorders are neurodegenerative.
Author Response
Manuscript ID: cimb-2478951.
Submission Title: Forced swimming-induced- depression and anxiety are reduced by chlorpheniramine by suppressing oxidative and inflammatory mediators and activating the Nrf2-BDNF signaling pathway.
Revised Title: Forced swimming induced depressive-like behavior and anxiety are reduced by chlorpheniramine via suppression of oxidative and inflammatory mediators and activating the Nrf2-BDNF signaling pathway
Authors: Hasan S. Alamri, Rana Mufti, Deema Kamal Sabir, Abdulwahab A. Abuderman, Amal F. Dawood, Asmaa M. ShamsEldeen, Mohamed A. Haidara, Esma R. Isenovic, and Mahmoud H El-Bidawy
We express our sincere gratitude for the work and time the Editor and Reviewers committed to helping us improve our manuscript. Following the reviewers' recommendation, we have corrected all pointed issues. We revised the manuscript's parts according to the reviewers' suggestions and provided the requested additional information. The changes in the manuscript text are highlighted in yellow. We believe that we have addressed all of the Reviewers' concerns, and we hope that the revised version of the manuscript is suitable for publication in the Current Issues in Molecular Biology.
RESPONSES TO THE EDITORS' COMMENTS
COMMENT. Please modify references in the text where reference numbers should be in square brackets and placed before punctuations.
RESPONSE: We appreciate the editor's input and apologize for deviating from the Journal's approach. In response to this comment, reference numbers are enclosed in square brackets and placed before punctuation in the updated version of the MS.
RESPONSES TO THE REVIEWERS' COMMENTS
Reviewer #2
COMMENT: The manuscript by Alamri et al., provides an important information on chlorpheniramine (CPA) that as it reported has antidepressant-like and anxiolytic-like activity in acutely stressed rats. The results are sounds,
RESPONSE: We thank the Reviewer for his encouraging comments.
Major comments
COMMENT 1a: Since the authors indicate the sample size as 6 animals in each group I have a serious concern regarding the statistical power that complicates the ability to draw appropriate interpretations from the results. Did the power analysis was performed?
RESPONSE: We thank the Reviewer for these valuable observations. Our decision to use only 6 rats per group stemmed from the principles of 3Rs (replacement, reduction, and refinement) and animal welfare (46). In addition, our previous published work has demonstrated that a minimum of 4 animals per group is sufficient to get apparent significant differences between two groups (control and treated). When planning our experiments, we have to take into consideration Ethical Considerations. We could not justify using more animals to increase our numbers when our data is already clearly significant, and these extra numbers would not alter our findings. We wished to use the fewest animals possible while enabling us to conclude with the desired confidence level. We have used sequential sampling to reduce the distress imposed on the animals.
Furthermore, based on the resource equation: E = N (number of animals per treatment x number of treatments) - T (number of treatments). Where N = the total number of subjects (e.g. individual animals or groups/cages of animals) and T = the number of treatment combinations, E (the sample size) should be between 10 and 20. In our experiment, we have only 1 treatment, using 4 rats per treatment: N = 4 (4 x 1) and T =1; therefore, E = 4-1 = 3. So, the outliers were not excluded. Furthermore, the power analysis shows that 6 animals are sufficient for in vivo investigations, which shows our estimates' accuracy and our study's ability to make conclusions (n=6).
Reference
Hubrecht RC, Carter E. The 3Rs and humane experimental technique: Implementing change. Animals 2019;9(10):754. doi: 10.3390/ ani9100754
COMMENT 1b: Degrees of freedom for ANOVAs are not indicated anywhere in the results section. So, the next question is: whether outliers have been excluded?
RESPONSE: We thank the Reviewer for this valuable observation. In response to this comment, we indicated the degrees of freedom for ANOVA in the revised version of the MS. In addition, we now provide details of ANOVA (F statistic and degrees of freedom (df)) in normally distributed variables analyzed by ANOVA. Please see the following Table below:
|
|
Sum of Squares |
df |
Mean Square |
F |
Overall P value |
|
|
MDA |
Between Groups |
31911.434 |
2 |
15955.717 |
73.322 |
<0.001 |
|
Within Groups |
3264.182 |
15 |
217.612 |
|
|
|
|
Total |
35175.616 |
17 |
|
|
|
|
|
SOD |
Between Groups |
151.919 |
2 |
75.959 |
65.939 |
<0.001 |
|
Within Groups |
17.280 |
15 |
1.152 |
|
|
|
|
Total |
169.198 |
17 |
|
|
|
|
|
IL-1 B |
Between Groups |
44693.818 |
2 |
22346.909 |
247.961 |
<0.001 |
|
Within Groups |
1351.840 |
15 |
90.123 |
|
|
|
|
Total |
46045.658 |
17 |
|
|
|
|
|
Noradrenaline |
Between Groups |
17965.671 |
2 |
8982.836 |
97.998 |
<0.001 |
|
Within Groups |
1374.947 |
15 |
91.663 |
|
|
|
|
Total |
19340.618 |
17 |
|
|
|
|
|
Cortisol |
Between Groups |
199.564 |
2 |
99.782 |
24.530 |
<0.001 |
|
Within Groups |
61.016 |
15 |
4.068 |
|
|
|
|
Total |
260.580 |
17 |
|
|
|
|
|
BDNF |
Between Groups |
123369.854 |
2 |
61684.927 |
111.713 |
<0.001 |
|
Within Groups |
8282.623 |
15 |
552.175 |
|
|
|
|
Total |
131652.478 |
17 |
|
|
|
|
|
No of line crossings (OPEN FIELD TEST) |
Between Groups |
3164.778 |
2 |
1582.389 |
52.109 |
<0.001 |
|
Within Groups |
455.500 |
15 |
30.367 |
|
|
|
|
Total |
3620.278 |
17 |
|
|
|
|
|
Time spent in open arms (S) (Elevated Plus Maze Test.) |
Between Groups |
2362.333 |
2 |
1181.167 |
155.190 |
<0.001 |
|
Within Groups |
114.167 |
15 |
7.611 |
|
|
|
|
Total |
2476.500 |
17 |
|
|
|
|
|
Head dipping (Elevated Plus Maze Test.) |
Between Groups |
296.333 |
2 |
148.167 |
43.016 |
<0.001 |
|
Within Groups |
51.667 |
15 |
3.444 |
|
|
|
|
Total |
348.000 |
17 |
|
|
|
|
|
Stretching (Elevated Plus Maze Test.) |
Between Groups |
50.333 |
2 |
25.167 |
31.027 |
<0.001 |
|
Within Groups |
12.167 |
15 |
0.811 |
|
|
|
|
Total |
62.500 |
17 |
|
|
|
|
COMMENT2. For comparison of two groups for immobility time the t-test is more suitable.
RESPONSE: We thank the Reviewer for this very valuable suggestion. We want to point out that we have readily used unpaired t-test for immobility time. We apologize for not clarifying clearly that a t-test was used to compare two groups. Please see the methodology section, where we added this important and previously missing information. The unpaired Student t-test was used to analyze differences between two groups in variable values that follow the normal distribution. The revised section is as follows:
Statistical Analysis
The data are expressed as mean ± standard deviation (SD). Data were processed and analyzed using GraphPad Prism (version 6). Data were double-checked for normality using the Shapiro-Wilk test and normality plots. The unpaired Student t-test was used to analyze differences between two groups in variable values that follow the normal distribution, while the Mann-Whitney test was used for non-normally distributed variables. One-way ANOVA followed by Tukey's post hoc test was used to analyze differences between three groups for normally distributed variables, while non-parametric Kruskal-Wallis was used for non-normally distributed variables. Pearson correlation statistical analysis was performed to analyze the correlation significance between two different parameters. Statistical significance was considered if p ≤ 0.05.
COMMENT 3. In the "materials and methods" section, the scheme of the experiment is required. So, the figure 8 is more appropriate in the paragraph 2.2. From the experimental design description it is not clear when exactly and in what sequence the OF and EPM tests were carried out. Were they carried out on days 16-20? From the description, I got the impression that both tests were performed on the same day. If this is true, then this is a significant drawback.
RESPONSE: We agree with the Reviewer's comment. In response to this comment, we shifted the graphical abstract to the "materials and methods" "section according to the ' 'Reviewer's comment (Now it becomes Figure 1., in the revised version of the MS.
COMMENT 4. From the experimental design description it is not clear when exactly and in what sequence the OF and EPM tests were carried out. Were they carried out on days 16-20?
RESPONSE: We thank the Reviewer for these observations and apologize for not being clear. All tests were done on day 16.
COMMENT 5. From the description, I got the impression that both tests were performed on the same day. If this is true, then this is a significant drawback.
RESPONSE: Again, the Reviewer's observation is correct. According to our previously published paper, all behavioral tests were conducted during a specific time frame (9 a.m.–1 p.m.), and each group was tested separately. Moreover, the order of the tested animal was chosen randomly within each group.
( Estaphan S, et al.), For assessment of acute changes, the animals must be examined on the same day, as previously conducted by McBride et al., 2017, authors assessed neurological and behavioral changes twenty-four hours post-injury using 17 cost-effective and easy-to-perform tests and developed neuroscore.
References:
Estaphan, S., Curpăn AS, Khalifa D, Rashed L, Ciobica A, Cantemir A, Ciobica A, Trus C, Ali M, ShamsEldeen A. Combined Low Dose of Ketamine and Social Isolation: A Possible Model of Induced Chronic Schizophrenia-Like Symptoms in Male Albino Rats. Brain Sci. 2021 July 11;11(7):917. doi: 10.3390/brainsci11070917. PMID: 34356151; PMCID: PMC8303272
McBride DW, Nowrangi D, Kaur H, Wu G, Huang L, Lekic T, Tang J, Zhang JH. A composite neurobehavioral test to evaluate acute functional deficits after cerebellar haemorrhage in rats. J Cereb Blood Flow Metab. 2018 Mar;38(3):433-446. doi: 10.1177/0271678X17696509. Epub 2017 March 20. PMID: 28318366; PMCID: PMC5851133.
COMMENT 6. Throughout the text, the authors use the term "depression". This is incorrect if we are talking about rodents. We can only use terms "depressive-like behavior" or"behavioral despair"..
RESPONSE: We agree with the Reviewer's comment. Corrected as suggested:
Before: depression
Now: depressive-like behavior
COMMENT 7. The authors assessed only BDNF mRNA level, but in fact, this may not reflect the changes in BDNF protein level. For example, proBDNF level may be increased, while the level of mature BDNF may not. So, this needs to be discussed and mentioned among the study limitation.
RESPONSE: We thank the Reviewer for these valuable suggestions. In response to this comment, we have discussed this concept regarding the study limitation. The newly added text now reads as follows:
"It is strongly suggested that more studies using a dose-response curve be conducted. Future research may be successful in identifying more pathways that control inflammatory biomarkers in various parts of the brain. Also, measurements of Nrf2 and BDNF utilizing Western blot to measure the actual protein level would be considered when we extend our research. Last but not least, the ability of CPA to guard against stress-related depressive-like behavior and anxiety was the main focus of this investigation. It could be more instructive to look at this impact on brain function and all other measured markers over a longer period. Additionally, only a potential protective effect of CPA against stress-related brain damage was demonstrated in this study. The ability of CPA to prevent stress-related harm to other organs, such as the heart, liver, lungs, and kidneys, was not examined in this study but will be in subsequent ones."
COMMENT 8). The statement "downregulation of the Nrf2-BDNF signaling pathway" in the discussion is too sounds. Actually only two components of this signaling pathway have been analyzed and each of them could changed independently. Considering a plethora of factors affecting BDNF expression, such a scenario is quite possible.
RESPONSE: We thank the Reviewer for this comment and are happy to hear that the Reviewer share our discussion regarding the Nrf2-BDNF signaling pathway.
Minor comments
COMMENT 1. I assume that the distribution was normal, which led to the choice of parametric ANOVA test, however, in the "statistical analysis" section, it is necessary to indicate whether a test for normality was carried out.
RESPONSE: We thank the Reviewer for this valuable observation. Data were -checked for normality using the Shapiro-Wilk test and normality plots, and most variables proved to be not deviated from normal distribution. In the statistical analysis section, we have indicated whether a test for normality was carried out.
COMMENT 2. In the figure legends as well as in the abstract the sample number indicated as n = 8. Obviously, this is mistake.
RESPONSE: We thank the Reviewer for this observation. We apologize for this mistake. In the revised version of the MS, we corrected it in the abstract, and corrections are made for all appropriate figures.
COMMENT 3. P values and indication on the graphs are inconsistent. For example, for immobility time indicated p ≤ 0.0001 while on the figure 1A placed (*) instead (***) as widely accepted for p ≤ 0.0001. Such discrepancies are throughout the text. Also I recommend to denote control vs. CPA+AS and AS vs. CPA+AS comparisons with different symbols (i.e. * and #). This should facilitate the visual perception of the material.
RESPONSE: We thank the Reviewer for these valuable suggestions. It was corrected as suggested.
Before: **
Now: #.
In addition, as we consider (*p or #p) significant compared to other groups, we added exact p values to all ligands in the text.
COMMENT 4. Subheading in the paragraph 3.6 obviously from any other paper.
RESPONSE: The Reviewer is right, and we apologize for this mistake. In the revised version of the MS, we changed the title to: "Correlation between anxiety-related behaviour and changes in brain transmitters, inflammatory biomarkers and Nrf2-BDNF signaling pathway".
COMMENT 5. Figure 2C: correctly as "number of episodes of grooming"
RESPONSE: We thank the Reviewer for this great suggestion. Corrected as suggested
Before: Grooming
Now: Number of episodes of grooming
COMMENT 6. The same researcher performed all behavioral tests" in the paragraph 2.3: what does it mean? This sentence needs to be made clear.
RESPONSE: We thank the Reviewer for his observation. We apologize for not being clear. The same researcher performed all behavioral tests…" means that the same person performed all behavioral tests.
COMMENT 7. The sentence "mental illnesses, including ' 'Alzheimer's and Parkinson's" is incorrect. Mentioned disorders are nurodegenerative
RESPONSE: Again, the Reviewer is right. It was corrected as suggested.
Before: mental illnesses
Now: neurodegenerative disorders.

Reviewer 3 Report
This paper reports on the ability of chlorpheniramine (CP) to modulate some biochemical parameters in rodents subjected to a forced swim test. In addition to the FST rodents were also tested in the open field and elevated plus maze. The sequence of the latter tests and whether randomisation of animals (by group i.e., control, acute stress (AS) or stress plus CP) to each of the tests is not reported. All tests were performed in the morning to avoid circadian effects (although the reason for the time of testing is not explicitly stated). The protocols for behavioural testing as outlined by the authors appear to have been those devised by the original instigators of such tests. The biochemical tests conducted were those which have been putatively associated with depression and anxiety states from either pre-clinical or clinical studies. Standard kits and protocols appear to have been followed.
There are several major issues which need to be addressed in this study:
1. The primary weakness of the study is that it employed only a single dose of CP. The dose chosen is not justified based on any previous findings of the authors or the literature. Thus, it is not possible to decide if CP demonstrates a dose-response relationship in the tests or if the dose response differs between tests or indeed of the effects on the biochemical parameters are dependent on the CP dose. The lack of dose response data is a significant weakness of the study and potentially clouds exploration of important pharmacological mechanisms involved in the results.
2. The results of the statistical tests fall short of the reporting standards expected for such tests. The F statistic and degrees of freedom for the ANOVA is not reported only the P value. It is not clear if the P values are for the omnibus ANOVA test or for the post-hoc tests. Whether the assumptions for applying ANOVA to the sets of data generated in the study do not appear to have been tested. For example, was Levene’s test (or some such equivalent) applied to test the equality of the variances? Where the data normally distributed, based on what test? If not then ANOVA is inappropriate and a non-parametric test should have been used.
3. The authors provide no indication of the power of the study based for example on the eta squared values from the ANOVA (notwithstanding the reservations about the appropriateness of the test as noted above). Given the small sample size for the behavioural tests such as described here, it seems likely that the power is low. Was a sample size calculation performed a priori?
Some general comments:
1. The FST is not an animal model of depression. It is a model which can be used to detect molecules which may possess antidepressant activity. In other words, it is a pharmacological predictive model, but it has significant translational shortcomings. Only a clinical study can decide if a molecule is an antidepressant or not. In other words, the FST only indicates if a molecule has ‘antidepressant-like’ activity.
2. It is surprising that the authors do not mention the relationship of CP to zimelidine (an SSRI antidepressant no longer in clinical use, but which was developed from CP by Carlson and colleagues in the 1980’s).
Some specific comments:
1. The authors present a confusing picture of the number of animals used in the study. The abstract says 18 rats and then talks of 3 groups of 8 (=24). Further in the captions to the figures n=8 is often reported. This needs clarification.
2. Abstract states animals were ‘slaughtered’. I think a preferred term would be ‘decapitated’ or ‘euthanised’.
3. P 2. Why refer to antipsychotic drugs when the study is about potential antidepressant activity?
4. P.2. The statement “Porsolt and his coworkers created FST to assess the therapeutic potential of antidepressant medications” seems about face. Rather the test was used to discover molecules with antidepressant potential. See the comments on animal tests above.
5. P2. References 30 and 31 do not refer in any way to the statements which they are purported to support here. In fact, this is a further weakness of the paper: many of the statements throughout the text are reputedly supported by references which are irrelevant to the statement at hand.
6. P.3. I believe what is meant is ‘anxiolytic’ or ‘anti-anxiogenic’ not ‘anti-anxiolytic’.
7. P.3. n=6 rats per group? see comment above.
8. P.3. Wistar not Wister.
9. P.3. Did control rats receive identical handling to "saline treated rats” including pain associated with i.p. injections?
10. P.3. Which university ethics committee approved the research as the authors are from different universities and countries?
11. P.3. What effect does sodium thiopentone have the biochemical measures used?
12. P.3. Were there other measures taken in the FST e.g., escape behaviour, swimming time? Maybe important in context of serotonergic / noradrenergic measures see Cryan et al Neurosci Biobehav Rev . 2005;29(4-5):547-69.
13. P.4. What was the intensity of light in lux at floor level for open field test?
14. P.5. Results are significant if P<0.05 not P>0.05.
15. P.5. Poor terminology used here: the rats are not depressed! The test predicts antidepressant-like activity!
16. P.5. While hypercortisolaemia may be true in the clinical state what you have investigated is an acute stress situation not depression per se.
17. P.5. two instances where p=0.xxxx simply state P NS, not significant.
18. P.8. The usual reporting for the elevated plus maze is to record separate open arm and closed arm entries. Open arm entries should increase for an anxiolytic.
19. P. 10. Title for section 3.6: What does this paper have to do with kidney damage?
20. P.11. What is the rationale for correlating the markers shown here? To correlate these measures, you need to suggest (show) that there is some a priori reason that mobility / immobility is in some caused by the biological markers measured. The mere correlation proves nothing without this justification and is simply random chance helped along by the dichotomous results. What happens if you include the data from the control animals? If the a priori relationship exists at all then including data from the controls make sense.
21. P.12. The statement “Our findings demonstrated that FST prolonged periods of immobility that indicated depression in our animal model (Fig. 1A)” really adds nothing new. The model id well known and is a pharmacological discovery model. (see comments above).
I do not have any major concerns with the English language usage. There are some specific uses which could be improved as the authors have used words which could be confusing.
Author Response
Manuscript ID: cimb-2478951.
Submission Title: Forced swimming-induced- depression and anxiety are reduced by chlorpheniramine by suppressing oxidative and inflammatory mediators and activating the Nrf2-BDNF signaling pathway.
Revised Title: Forced swimming induced depressive-like behavior and anxiety are reduced by chlorpheniramine via suppression of oxidative and inflammatory mediators and activating the Nrf2-BDNF signaling pathway
Authors: Hasan S. Alamri, Rana Mufti, Deema Kamal Sabir, Abdulwahab A. Abuderman, Amal F. Dawood, Asmaa M. ShamsEldeen, Mohamed A. Haidara, Esma R. Isenovic and Mahmoud H El-Bidawy
We express our sincere gratitude for the work and time the Editor and Reviewers committed to helping us improve our manuscript. Following the reviewers' recommendation, we have corrected all pointed issues. We revised the manuscript's parts according to the reviewers' suggestions and provided the requested additional information. The changes in the manuscript text are highlighted in yellow. We believe that we have addressed all of the Reviewers' concerns, and we hope that the revised version of the manuscript is suitable for publication in the Current Issues in Molecular Biology.
RESPONSES TO THE REVIEWERS' COMMENTS
Reviewer #3
COMMENT 1. This paper reports on the ability of chlorpheniramine (CP) to modulate some biochemical parameters in rodents subjected to a forced swim test. In addition to the FST rodents were also tested in the open field and elevated plus maze.
RESPONSE: We appreciate the Reviewer's comments summarizing our research's main points.
COMMENT 2. The sequence of the latter tests and whether randomization of animals (by group i.e., control, acute stress (AS) or stress plus CP) to each of the tests is not reported.
RESPONSE: We thank the Reviewer for this question and apologize if we were unclear. To make clear what was done, we have corrected the section methodology. Moreover, the corrected text now reads as follows:
2.2. Experimental Design
After a week of acclimation, 18 Wistar rats were evenly divided into three groups (n = 6 rats in each group). Three groups of animals were separated. Group 1 contains control normal rats receiving saline (i.p.) pretreatment for 2 weeks, Group 2 contains rats that underwent a forced swimming stress test after receiving saline (i.p.) pretreatment for 2 weeks, and Group 3 contains rats that underwent a forced swimming stress test after receiving CPA dissolved in saline (i.p.) at a dose of 10 mg/kg for 2 weeks after (Hirano et al., 2006). All experimental protocols adhered to the guidelines of the Animal Welfare Committee of the Universidad Complutense following European legislation (2010/63/EU). The current study was approved by the 6th October University Research Ethical Committee; 06U REC, (NO: PRE-Me-2103018).
All animals underwent behavioral tests after the experiment. Next, rats were given sodium phenobarbital anaesthesia (40 mg/kg body weight), and blood was drawn via cardiac puncture into plain tubes for serum separation and cortisol measurement. (schematic presentation shows all experimental steps that were done, Figure 1)
The animals were subsequently sacrificed by decapitation, and the brain tissues were harvested to assess the hippocampus MDA, SOD, IL-1β, and IL-6. Serotonin, dopamine, Nrf2, and BDNF levels were assessed in the cerebral cortex. At -80 degrees, brain tissue and serum were both preserved.
2.3. Behavioral Tests
The same researcher performed all behavioral tests (Savage et al.) between 8:00 a.m. and 1:00 p.m. by the same researcher. Rats were given 30 minutes to adjust to the test environment before being tested. According to our previously published paper, all behavioral tests were conducted during a specific time frame, and each group was tested separately. Moreover, the order of the tested animal was chosen randomly within each group (Estaphan et al., 2021).
COMMENT 2.: All tests were performed in the morning to avoid circadian effects (although the reason for the time of testing is not explicitly stated). The protocols for behavioural testing as outlined by the authors appear to have been those devised by the original invstigators of such tests. The biochemical tests conducted were those which have been putatively associated with depression and anxiety states from either pre-clinical or clinical studies. Standard kits and protocols appear to have been followed.
RESPONSE: We thank the Reviewer for encouraging comments.
Major issues which need to be addressed in this study:
There are several major issues which need to be addressed in this study:
COMMENT 1: The primary weakness of the study is that it employed only a single dose of CP. The dose chosen is not justified based on any previous findings of the authors or the literature. Thus, it is not possible to decide if CP demonstrates a dose-response relationship in the tests or if the dose response differs between tests or indeed of the effects on the biochemical parameters are dependent on the CP dose. The lack of dose response data is a significant weakness of the study and potentially clouds exploration of important pharmacological mechanisms involved in the results.
RESPONSE: Thanks for the Reviewer's comments. We follow the previous work of (Hirano et al., 2006), showed that the histamine H(1) receptor antagonist chlorpheniramine (1-10 mg/kg, s.c.) dose-dependently and significantly reduced the duration of immobility in both non-diabetic and diabetic mice. So we chose to use the highest dose. We regret that a different reference was accidentally added, and we changed it to Hirano et al., 2006.
Reference:
HIRANO, S., MIYATA, S., ONODERA, K. & KAMEI, J. 2006. Effects of histamine H(1) receptor antagonists on depressive-like behavior in diabetic mice. Pharmacol Biochem Behav, 83, 214-20.
COMMENT 2: The results of the statistical tests fall short of the reporting standards expected for such tests. The F statistic and degrees of freedom for the ANOVA is not reported only the P value. It is not clear if the P values are for the omnibus ANOVA test or for the post-hoc tests. Whether the assumptions for applying ANOVA to the sets of data generated in the study do not appear to have been tested. For example, was ' 'Levene's test (or some such equivalent) applied to test the equality of the variances? Where the data normally distributed, based on what test? If not then ANOVA is inappropriate and a non-parametric test should have been used.
RESPONSE: We apologize for not making the statistical analyses obvious. The Table below provides an overview of ANOVA (F statistic and degrees of freedom (df)) for normally distributed variables. The p values are related to F values. The Shapiro-Wilk test is used to determine whether data distribution is normal.
|
|
Sum of Squares |
df |
Mean Square |
F |
Overall P value |
|
|
MDA |
Between Groups |
31911.434 |
2 |
15955.717 |
73.322 |
<0.001 |
|
Within Groups |
3264.182 |
15 |
217.612 |
|
|
|
|
Total |
35175.616 |
17 |
|
|
|
|
|
SOD |
Between Groups |
151.919 |
2 |
75.959 |
65.939 |
<0.001 |
|
Within Groups |
17.280 |
15 |
1.152 |
|
|
|
|
Total |
169.198 |
17 |
|
|
|
|
|
IL-1 B |
Between Groups |
44693.818 |
2 |
22346.909 |
247.961 |
<0.001 |
|
Within Groups |
1351.840 |
15 |
90.123 |
|
|
|
|
Total |
46045.658 |
17 |
|
|
|
|
|
Noradrenaline |
Between Groups |
17965.671 |
2 |
8982.836 |
97.998 |
<0.001 |
|
Within Groups |
1374.947 |
15 |
91.663 |
|
|
|
|
Total |
19340.618 |
17 |
|
|
|
|
|
Cortisol |
Between Groups |
199.564 |
2 |
99.782 |
24.530 |
<0.001 |
|
Within Groups |
61.016 |
15 |
4.068 |
|
|
|
|
Total |
260.580 |
17 |
|
|
|
|
|
BDNF |
Between Groups |
123369.854 |
2 |
61684.927 |
111.713 |
<0.001 |
|
Within Groups |
8282.623 |
15 |
552.175 |
|
|
|
|
Total |
131652.478 |
17 |
|
|
|
|
|
No of line crossings (OPEN FIELD TEST) |
Between Groups |
3164.778 |
2 |
1582.389 |
52.109 |
<0.001 |
|
Within Groups |
455.500 |
15 |
30.367 |
|
|
|
|
Total |
3620.278 |
17 |
|
|
|
|
|
Time spent in open arms (S) (Elevated Plus Maze Test.) |
Between Groups |
2362.333 |
2 |
1181.167 |
155.190 |
<0.001 |
|
Within Groups |
114.167 |
15 |
7.611 |
|
|
|
|
Total |
2476.500 |
17 |
|
|
|
|
|
Head dipping (Elevated Plus Maze Test.) |
Between Groups |
296.333 |
2 |
148.167 |
43.016 |
<0.001 |
|
Within Groups |
51.667 |
15 |
3.444 |
|
|
|
|
Total |
348.000 |
17 |
|
|
|
|
|
Stretching (Elevated Plus Maze Test.) |
Between Groups |
50.333 |
2 |
25.167 |
31.027 |
<0.001 |
|
Within Groups |
12.167 |
15 |
0.811 |
|
|
|
|
Total |
62.500 |
17 |
|
|
|
|
COMMENT 3:The authors provide no indication of the power of the study based for example on the eta squared values from the ANOVA (notwithstanding the reservations about the appropriateness of the test as noted above). Given the small sample size for the behavioural tests such as those described here, it seems likely that the power is low. Was a sample size calculation performed a priori?
RESPONSE: Considering that the study is pilot, experimental, performed on animals, and in line with our responses to Reviewer 2 Comment# 1, we make an estimated sample size for performed study according to Power analysis and Sample size software (www.ncss.com).
Some general comments:
COMMENT4: The FST is not an animal model of depression. It is a model which can be used to detect molecules which may possess antidepressant activity. In other words, it is a pharmacological predictive model, but it has significant translational shortcomings. Only a clinical study can decide if a molecule is an antidepressant or not. In other words, the FST only indicates if a molecule has 'antidepressant-'like' activity.
RESPONSE: We thank the Reviewer for the comment. We based our work on (Yankelevitch-Yahav et al., 2015), who documented that FST can represent rodent depression like-behaviour. However, we will add this to the limitation of the study.
COMMENT5: It is surprising that the authors do not mention the relationship of CP to zimelidine (an SSRI antidepressant no longer in clinical use, but which was developed from CP by Carlson and colleagues in the 1980's).
RESPONSE: We thank the Reviewer for his valuable comment. We 'did not mention that Zimelidine has been banned worldwide due to severe, sometimes fatal, cases of central and/or peripheral neuropathy known as Guillain-Barré syndrome and due to a peculiar hypersensitivity reaction including arthralgia. Additionally, Zimelidine was charged with causing an increase in suicidal ideation and/or attempts among depressive patients.
Some specific comments:
COMMENT 6:The authors present a confusing picture of the number of animals used in the study. The abstract says 18 rats and then talks of 3 groups of 8 (=24). Further in the captions to the figures n=8 is often reported. This needs clarification.
RESPONSE: Thanks for the Reviewer's comment. We apologize for this mistake and fixed it to be (n=6) all over the text and ligands.
COMMENT 7:Abstract states animals were ''slaughtered'. I think a preferred term would be '''decapitated' or ''euthanized'.
RESPONSE: Thanks for the Reviewer's comment. We changed it according to the valuable comments of the Reviewer.
Before: slaughtered
Now: euthanized
COMMENT 8 P.2.: Why refer to antipsychotic drugs when the study is about potential antidepressant activity?
RESPONSE: We thank the Reviewer for his comment. Furthermore, we change the sentence to" Animal models of behavioral and/or emotional disorders become a very alluring subject for evaluating anxiety-like (open field test, novelty suppressed feeding, elevated plus maze, light/dark box, stress-induced hyperthermia) and depression-like behaviors "(Belovicova et al., 2017)".
Reference:
BELOVICOVA, K., BOGI, E., CSATLOSOVA, K. & DUBOVICKY, M. 2017. Animal tests for anxiety-like and depression-like behavior in rats. Interdiscip Toxicol, 10, 40-43.
COMMENT 9 P.2.: The statement "Porsolt and his coworkers created FST to assess the therapeutic potential of antidepressant medications" seems about-face.? Rather the test was used to discover molecules with antidepressant potential. See the comments on animal tests above.
RESPONSE: We thank the Reviewer for comment. As stated above, we based our work on (Yankelevitch-Yahav et al., 2015), who documented that FST can represent rodent depression like-behavior. However, we will add this to the limitation of the study.
COMMENT10 P2: References 30 and 31 do not refer in any way to the statements which they are purported to support here. In fact, this is a further weakness of the paper: many of the statements throughout the text are reputedly supported by references which are irrelevant to the statement at hand.
RESPONSE: We apologize for this mistake. We deleted N0 30 and replaced it. However, in Ref 31, we changed the sentence to fit the reference.
COMMENT 11 P.3. :I believe what is meant is '''anxiolytic' or 'anti-'anxiogenic' not 'anti-'anxiolytic'.
RESPONSE: We agree with the Reviewer's comments. It was corrected as suggested.
Before: anti-anxiolytic
Now: anti-anxiogenic
COMMENT 12. P.3.: n=6 rats per group? see comment above.
RESPONSE: We apologize for any typos. Within the full MS, we made the necessary corrections.
COMMENT 13 P.3.: Wistar not Wister.
RESPONSE: We apologize for any typos. Within the full MS, we made the necessary corrections.
Before: Wister.
Now: Wistar
COMMENT 14. P.3. : Did control rats receive identical handling to "saline-treated rats"""" including pain associated with i.p. injections?
RESPONSE: We appreciate the Reviewer making this critical point. The answer is Yes. Saline was given i.p.. to the control animals as a vehicle. This information was added to the Methodology section.
COMMENT 15. P.3.: Which university ethics committee approved the research as the authors are from different universities and countries?
RESPONSE: On October 6t, 2021, University Research Ethical Committee, 06U REC (NO: PRE-Me-2103018), approved the current study. The method section now includes this statement.
COMMENT 16 P.3. :What effect does sodium thiopentone have on the biochemical measures used?
RESPONSE: We apologize for this mistake. It was written by mistake that we used thiopental. Rats were sacrificed (using sodium phenobarbital at a dose of 40 mg/kg i.p., rats were anaesthetized, and blood was drawn via cardiac puncture into plain tubes for serum separation and cortisol measurement, then were killed by decapitation as we did in our previous research (Estaphan et al., 2021).
COMMENT 17 P.3.: Were there other measures taken in the FST e.g., escape behaviour, swimming time? Maybe important in the context of serotonergic / noradrenergic measures see Cryan et al Neurosci Biobehav Rev . 2005;29(4-5):547-69.
RESPONSE: We thank the Reviewer for this valuable comment and add this to our manuscript discussion section as a method that should be utilized rather than FSW. (However, Cryan et al. recommended that the modified rat FST become accepted as an improved method for characterizing the effects of antidepressant drugs and studying the neural substrates underlying their behavioral effects(Cryan et al., 2005).
COMMENT 18 P.4. :What was the intensity of light in lux at floor level for open field test?
RESPONSE: We thank the Reviewer for this helpful question. The centre area was illuminated to 100 lx by an LED light source attached to the LED light source above the arena's centre. We added this information in the methodology section.
COMMENT 19. P.5. :Results are significant if P<0.05 not P>0.05.
RESPONSE: We apologize for this typo mistake, and this was corrected
COMMENT 20P.5. Poor terminology used here: the rats are not depressed! The test predicts antidepressant-like activity!
RESPONSE: The Reviewer's observation is correct. In response to this comment, we corrected it all over the manuscript, even in the title.
COMMENT 21. P.5. :While hypercortisolaemia may be true in the clinical state what you have investigated is an acute stress situation not depression per se.
RESPONSE: Thanks for the Reviewer's comment. However, we measure it in the context that FST induces depressive -like behaviour. We found that it increased, as our results showed (Fig 2B)
COMMENT 22 P.5. :two instances where p=0.xxxx simply state P NS, not significant.
RESPONSE: Thanks for this statement; we corrected it to not significant.
COMMENT 23. P.8.: The usual reporting for the elevated plus maze is to record separate open arm and closed arm entries. Open arm entries should increase for an anxiolytic.
RESPONSE: We agree with the Reviwer's comment. Our results follow the former poiner study of Pellow et al. (1985). Authors reported significantly fewer entries into the open arms than into the closed arms and spent significantly less time in open arms. Confinement to open arms was associated with the observation of significantly more anxiety-related behaviours.
Reference:
Pellow S, Chopin P, File SE, Briley M. Validation of open:closed arm entries in an elevated plus-maze as a measure of anxiety in the rat. J Neurosci Methods. 1985 Aug;14(3):149-67. doi: 10.1016/0165-0270(85)90031-7. PMID: 2864480.
COMMENT24 P. 10. :Title for section 3.6: What does this paper have to do with kidney damage?
RESPONSE: Thanks for the Reviewer's comment. We apologize for this mistake, and it was changed to 3.6. Correlation between depression-like behaviour and the number of line crossings
COMMENT 25 P.11.: What is the rationale for correlating the markers shown here? To correlate these measures, you need to suggest (show) that there is some a priori reason that mobility / immobility is in some caused by the biological markers measured. The mere correlation proves nothing without this justification and is simply random chance helped along by the dichotomous results. What happens if you include the data from the control animals? If the a priori relationship exists at all then including data from the controls make sense.
RESPONSE: We thank the Reviewer for his comment. As we agree with Reviewer's comment we deleted this correlation and removed Fig 7. Now Fig 8 becomes Fig 7
COMMENT 26 P.12. : The statement "Our findings demonstrated that FST prolonged periods of immobility that indicated depression in our animal model (Fig. 1A")" really adds nothing new. The model is well-known and is a pharmacological discovery model. (see comments above).
RESPONSE: We appreciate the Reviewer's input. In response to these comments, we changed it to "Our results indicated depressive-like behavior in our animal model as it was associated with increased mobility time (Fig. 2A)." However, Cryan et al. (2005) suggested that the modified rat FST has become regarded as a superior tool for evaluating the effects of antidepressant medications as well as examining the brain substrates underlying their behavioral effects.
Comments on the Quality of English Language
COMMENT 27: I do not have any major concerns with the English language usage. There are some specific uses which could be improved as the authors have used words which could be confusing.
RESPONSE: We thank the Reviewer for this comment. In response to this comment, we did English editing, and to improve the text of the MS, we changed the confusing words.

Round 2
Reviewer 2 Report
The most of comments adequately addressed. The details of ANOVA (F statistic and degrees of freedom (df)) are presented only in the response to comments but not in the MS. These detail should be included in the MS text
Author Response
Manuscript ID: cimb-2478951.
Submission Title: Forced swimming-induced- depression and anxiety are reduced by chlorpheniramine by suppressing oxidative and inflammatory mediators and activating the Nrf2-BDNF signaling pathway.
Revised Title: Forced swimming induced depressive-like behavior and anxiety are reduced by chlorpheniramine via suppression of oxidative and inflammatory mediators and activating the Nrf2-BDNF signaling pathway
Authors: Hasan S. Alamri, Rana Mufti, Deema Kamal Sabir, Abdulwahab A. Abuderman, Amal F. Dawood, Asmaa M. Shams Eldeen, Mohamed A. Haidara, Esma R. Isenovic and Mahmoud H El-Bidawy
We express our sincere gratitude for the work and time the Editor and Reviewers committed to helping us improve our manuscript. Following the reviewers' recommendation, we have corrected all pointed issues. We revised the manuscript's parts according to the reviewers' suggestions and provided the requested additional information. The changes in the manuscript text are highlighted in yellow. We believe that we have addressed all of the Reviewers' concerns, and we hope that the revised version of the manuscript is suitable for publication in the Current Issues in Molecular Biology.
RESPONSES TO THE REVIEWERS' COMMENTS
Reviewer # 2
COMMENT 1: Most of the comments were adequately addressed.
RESPONSE: We thank the Reviewer for accepting our response to the Reviewer's valuable comments.
COMMENT 2: The details of ANOVA (F statistic and degrees of freedom (df)) are presented only in response to comments but not in the MS. These detail should be included in the MS text
RESPONSE: We appreciate the Reviewer's supportive remarks. The ANOVA details (F statistic and degrees of freedom (df)) are now also provided in the revised version of the MS. In response to this comment, we now provide details of ANOVA (F statistic and degrees of freedom (df)) in normally distributed variables analyzed by ANOVA. The Table below is included in the revised version of the MS.
|
|
Sum of Squares |
df |
Mean Square |
F |
Overall P value |
|
|
MDA |
Between Groups |
31911.434 |
2 |
15955.717 |
73.322 |
<0.001 |
|
Within Groups |
3264.182 |
15 |
217.612 |
|
|
|
|
Total |
35175.616 |
17 |
|
|
|
|
|
SOD |
Between Groups |
151.919 |
2 |
75.959 |
65.939 |
<0.001 |
|
Within Groups |
17.280 |
15 |
1.152 |
|
|
|
|
Total |
169.198 |
17 |
|
|
|
|
|
IL-1 B |
Between Groups |
44693.818 |
2 |
22346.909 |
247.961 |
<0.001 |
|
Within Groups |
1351.840 |
15 |
90.123 |
|
|
|
|
Total |
46045.658 |
17 |
|
|
|
|
|
Noradrenaline |
Between Groups |
17965.671 |
2 |
8982.836 |
97.998 |
<0.001 |
|
Within Groups |
1374.947 |
15 |
91.663 |
|
|
|
|
Total |
19340.618 |
17 |
|
|
|
|
|
Cortisol |
Between Groups |
199.564 |
2 |
99.782 |
24.530 |
<0.001 |
|
Within Groups |
61.016 |
15 |
4.068 |
|
|
|
|
Total |
260.580 |
17 |
|
|
|
|
|
BDNF |
Between Groups |
123369.854 |
2 |
61684.927 |
111.713 |
<0.001 |
|
Within Groups |
8282.623 |
15 |
552.175 |
|
|
|
|
Total |
131652.478 |
17 |
|
|
|
|
|
No of line crossings (OPEN FIELD TEST) |
Between Groups |
3164.778 |
2 |
1582.389 |
52.109 |
<0.001 |
|
Within Groups |
455.500 |
15 |
30.367 |
|
|
|
|
Total |
3620.278 |
17 |
|
|
|
|
|
Time spent in open arms (S) (Elevated Plus Maze Test.) |
Between Groups |
2362.333 |
2 |
1181.167 |
155.190 |
<0.001 |
|
Within Groups |
114.167 |
15 |
7.611 |
|
|
|
|
Total |
2476.500 |
17 |
|
|
|
|
|
Head dipping (Elevated Plus Maze Test.) |
Between Groups |
296.333 |
2 |
148.167 |
43.016 |
<0.001 |
|
Within Groups |
51.667 |
15 |
3.444 |
|
|
|
|
Total |
348.000 |
17 |
|
|
|
|
|
Stretching (Elevated Plus Maze Test.) |
Between Groups |
50.333 |
2 |
25.167 |
31.027 |
<0.001 |
|
Within Groups |
12.167 |
15 |
0.811 |
|
|
|
|
Total |
62.500 |
17 |
|
|
|
|
Table 1. The ANOVA details (F statistic and degrees of freedom (df))
Reviewer 3 Report
It is not clear what is the relevance of the additional (from the original version) paragraphs on NO and VEGF to the current experiments. There does not appear to be any further use made of these ideas in the discussion, nor are any of these measured as part of the current study. In these paragraphs appears the statement that stress is d with low VEGF levels. The sentence does not make any sense. Further in these paragraphs CBX is introduced without definition.
Displayed results appears later in the introduction. What is meant by this: published results?
It was discovered that CPA (....?) antidepressant roles in a mouse model of anxiety. This sentence seems to be a non-sequitur: how can the drug have antidepressant roles(?) in a model of anxiety? What do you mean by antidepressant roles? The sentence has no verb.
"anti-anxiogenic" surely a better word is "anxiolytic"?
In the "Limitations of Study" the authors should include both sample size (the study has small numbers and is likely unpowered. Was there a power analysis?) and lack of a demonstrable dose response relationship (in this study). Further the sentences: In other words, the FST only indicates if a molecule has an 'antidepressant-liker. When we extend our research, we will induce stress and then consider FST as a test though [40] documented that FST can be a representation of rodent behavior-like behavior do not make sense.
What is rodent behaviour-like behaviour?
What is antidepressant liker? Do you mean antidepressant-like behavioural effects?
Some minor issues with one or two sentences in the text with obscure meanings.
Author Response
Manuscript ID: cimb-2478951.
Submission Title: Forced swimming-induced- depression and anxiety are reduced by chlorpheniramine by suppressing oxidative and inflammatory mediators and activating the Nrf2-BDNF signaling pathway.
Revised Title: Forced swimming induced depressive-like behavior and anxiety are reduced by chlorpheniramine via suppression of oxidative and inflammatory mediators and activating the Nrf2-BDNF signaling pathway
Authors: Hasan S. Alamri, Rana Mufti, Deema Kamal Sabir, Abdulwahab A. Abuderman, Amal F. Dawood, Asmaa M. Shams Eldeen, Mohamed A. Haidara, Esma R. Isenovic and Mahmoud H El-Bidawy
We express our sincere gratitude for the work and time the Editor and Reviewers committed to helping us improve our manuscript. Following the reviewers' recommendation, we have corrected all pointed issues. We revised the manuscript's parts according to the reviewers' suggestions and provided the requested additional information. The changes in the manuscript text are highlighted in yellow. We believe that we have addressed all of the Reviewers' concerns, and we hope that the revised version of the manuscript is suitable for publication in the Current Issues in Molecular Biology.
RESPONSES TO THE REVIEWERS' COMMENTS
Reviewer # 3
COMMENT 1: It is not clear what is the relevance of the additional (from the original version) paragraphs on NO and VEGF to the current experiments. There does not appear to be any further use made of these ideas in the discussion, nor are any of these measured as part of the current study. In these paragraphs appears the statement that stress is with low VEGF levels. The sentence does not make any sense. Further in these paragraphs, CBX is introduced without definition.
RESPONSE: We thank the Reviewer for these valuable observations. We have included the additional text as a response to the previous comment of Reviewer #1. In the revised version of MS, we added a new text under the section Study limitations which now reads as follows:
"Endothelial dysfunction (ED) has been connected with a variety of clinical disorders including depression and cardiovascular risk [56-58]. Furthermore, bipolar depression and increased vascular endothelial growth factor (VEGF) levels have frequently been linked [59]. The profile of circulating endothelium damage markers identified in the forced swimming-induced behavioral impairment and the relationship between the behavioral effects of CPA and nitric oxide (NO) and VEGF levels should investigated in further studies."
References:
- Giuseppe Murdaca, Francesca Spanò, Paola Cagnati & Francesco Puppo (2013) Free radicals and endothelial dysfunction: Potential positive effects of TNF-α inhibitors, Redox Report, 18:3, 95-99, DOI: 10.1179/1351000213Y.0000000046
- 57. Sherwood A, Hinderliter AL, Watkins LL, Waugh RA, Blumenthal JA. Impaired endothelial function in coronary heart disease patients with depressive symptomatology. J Am Coll Cardiol 2005; 46: 656–659. [PubMed] [Google Scholar]
- Rajendran P, Rengarajan T, Thangavel J, Nishigaki Y, Sakthisekaran D, Sethi G et al. The vascular endothelium and human diseases. Int J Biol Sci 2013; 9: 1057–1069. [PMC free article] [PubMed] [Google Scholar]
- Cooper DC, Tomfohr LM, Milic MS, Natarajan L, Bardwell WA, Ziegler MG et al. Depressed mood and flow-mediated dilation: a systematic review and meta-analysis. Psychosom Med 2011; 73: 360–369. [PMC free article] [PubMed] [Google Scholar]
COMMENT 2: Displayed results appear later in the Introduction. What is meant by this: published results?
RESPONSE: We thank the Reviewer for these valuable observations. We changed the sentence to "Data showed that …… ".
COMMENT 3: It was discovered that CPA (....?) antidepressant roles in a mouse model of anxiety. This sentence seems to be a non-sequitur: how can the drug have antidepressant roles(?) in a model of anxiety? What do you mean by antidepressant roles? The sentence has no verb.
RESPONSE: We thank the Reviewer for this valuable observation. In response to this comment, we changed the sentence to "It was discovered that CPA has an antidepressant effect in a mouse model of anxiety."
COMMENT4: anti-anxiogenic" surely a better word is "anxiolytic"?
RESPONSE: We thank the Reviewer for this valuable suggestion. In response to this comment, we changed anti-anxiogenic" to anxiolytic.,
COMMENT 5. In the "Limitations of Study" the authors should include both sample sizes (the study has small numbers and is likely unpowered. Was there a power analysis?
RESPONSE: We thank the Reviewer for this very valuable suggestion. The power analysis shows that 6 animals are sufficient for in vivo investigations, which shows our estimates' accuracy and our study's ability to make conclusions (n=6). Considering that the study is pilot, experimental, and performed on animals, we make an estimated sample size for performed study according to Power analysis and Sample size software (www.ncss.com).
COMMENT 6. A lack of a demonstrable dose-response relationship (in this study). Further the sentences: In other words, the FST only indicates if a molecule has an 'antidepressant-liker. When we extend our research, we will induce stress and then consider FST as a test though [40] documented that FST can be a representation of rodent behavior-like behavior does not make sense.
RESPONSE: We thank the Reviewer for these observations and apologize for not being clear. In response to this comment, we changed this sentence to "When we extend our research, we will induce stress and then consider FST as a test, not as an inducer of depression. Please see the Study Limitation section in the revised MS.
COMMENT 7. What is rodent behaviour-like behaviour?
RESPONSE: Again, the Reviewer's observation is correct. We deleted this sentence. Please see the Study Limitation section in the text of the revised version of the MS.
COMMENT 8. What is antidepressant liker? Do you mean antidepressant-like behavioural effects?
RESPONSE: We agree with the Reviewer's comment. Corrected as suggested:
Before: antidepressant-liker
Now: antidepressant effect
Comments on the Quality of English Language
COMMENT 9: Some minor issues with one or two sentences in the text with obscure meanings.
RESPONSE: We thank the Reviewer for this comment. In response to this comment, we did English editing, and to improve the text of the MS, we changed the confusing words.